# DMDSPEECH: DISTILLED DIFFUSION MODEL SURPASSING THE TEACHER IN ZERO-SHOT SPEECH SYNTHESIS VIA DIRECT METRIC OPTIMIZATION

## ABSTRACT

Diffusion models have demonstrated significant potential in speech synthesis tasks, including text-to-speech (TTS) and voice cloning. However, their iterative denoising processes are inefficient and hinder the application of end-to-end optimization with perceptual metrics. In this paper, we propose a novel method of distilling TTS diffusion models with direct end-to-end evaluation metric optimization, achieving state-of-the-art performance. By incorporating Connectionist Temporal Classification (CTC) loss and Speaker Verification (SV) loss, our approach optimizes perceptual evaluation metrics, leading to notable improvements in word error rate and speaker similarity. Our experiments show that DMDSpeech consistently surpasses prior state-of-the-art models in both naturalness and speaker similarity while being significantly faster. Moreover, our synthetic speech has a higher level of voice similarity to the prompt than the ground truth in both human evaluation and objective speaker similarity metric. This work highlights the potential of direct metric optimization in speech synthesis, allowing models to better align with human auditory preferences. The audio samples are available at `https://dmdspeech.github.io/demo/`

## 1 INTRODUCTION

Text-to-speech (TTS) technology has witnessed remarkable progress over the past few years, achieving near-human or even superhuman performance on various benchmark datasets (Tan et al., 2024; Li et al., 2024a; Ju et al., 2024). With the rise of large language models (LLMs) and scaling law (Kaplan et al., 2020), the focus of TTS research has shifted from small-scale datasets to large-scale models trained on tens to hundreds of thousands of hours of data encompassing a wide variety of speakers (Wang et al., 2023a;c; Shen et al., 2024; Peng et al., 2024; Łajszczak et al., 2024; Li et al., 2024b). Two primary methodologies have emerged for training these large-scale models: diffusion-based approaches and autoregressive language modeling (LM)-based methods. Both frameworks enable end-to-end speech generation without the need for hand-engineered features such as prosody and duration modeling as seen in works before the LLM era (Ren et al., 2020; Kim et al., 2021), simplifying the TTS pipeline and improving scalability.

Diffusion-based speech synthesis models have demonstrated superior robustness compared to LM-based approaches (Le et al., 2024; Lee et al., 2024). By generating all speech tokens simultaneously rather than sequentially, diffusion models avoid the error accumulation inherent in autoregressive models and offer faster generation times for longer sentences since their inference speed is not directly proportional to speech length (Ju et al., 2024). However, a significant drawback of diffusion models is their reliance on iterative sampling processes, which can be computationally intensive and time-consuming (Liu et al., 2024). Additionally, unlike LM-based models that directly estimate the likelihood of the output token distribution, diffusion models focus on estimating the score function rather than generating speech tokens end-to-end (Yang et al., 2023). This characteristic makes it inconvenient to optimize directly on the target distribution space using techniques such as perceptual loss (Johnson et al., 2016) or reinforcement learning with human feedback (RLHF) (Bai et al., 2022), as the models do not produce explicit token probabilities that can be easily adjusted.

To address these limitations, diffusion distillation techniques have been proposed to reduce sampling time (Salimans & Ho, 2022; Song et al., 2023; Sauer et al., 2023; Yin et al., 2024b). By distilling the diffusion model into a one-step generator, it becomes possible to access the final output directly and apply metric optimization methods to improve the desired attributes in the synthesized speech.

In this work, we introduce **DMDSpeech**, a distilled diffusion-based speech synthesis model designed for efficient and high-quality zero-shot speech generation. By employing distribution matching distillation (DMD) (Yin et al., 2024b;a), we transform the diffusion teacher model into a distilled student model that generates speech in 4 steps. The distilled model provides a direct gradient pathway between the noise input and the speech output, enabling end-to-end optimization of any differentiable metrics. We leverage two metrics relevant to zero-shot speech synthesis: speaker similarity via speaker verification (SV) loss and intelligibility and robustness via connectionist temporal classification (CTC) loss. With the SV loss, we enhance the model's ability to reproduce the unique characteristics of the prompt speaker, obtaining higher similarity to the prompt judged by both human evaluations and speaker verification models. Additionally, the CTC loss helps improve word error rate (WER), enhancing the intelligibility and accuracy of synthesized speech and achieving lower WER than ground truth and other end-to-end baseline models.

Our contributions are threefold:

- We present a distilled diffusion model for zero-shot speech synthesis that achieves state-of-the-art performance with significantly reduced inference time.
- We introduce a new speech synthesis framework for end-to-end optimization of perceptual metrics. We demonstrate that optimizing SV loss and CTC loss leads to improvements in speaker similarity and WER, respectively.
- We present an in-depth analysis of our model's ablations of DMD and direct metric optimization, demonstrating the correlations between human evaluations and perceptual metrics, while highlighting the trade-off between sampling speed and diversity (mode shrinkage).

## 2 RELATED WORKS

**Zero-Shot Text-to-Speech Synthesis**    Zero-shot TTS generates speech in an unseen speaker's voice using a small reference sample without additional training. Initial methods relied on speaker embeddings from pre-trained encoders (Casanova et al., 2022; 2021; Wu et al., 2022; Lee et al., 2022) or end-to-end speaker encoders (Li et al., 2024a; Min et al., 2021; Li et al., 2022; Choi et al., 2022), which required extensive feature engineering and struggled with generalization, limiting scalability. More recently, prompt-based methods using in-context learning with reference prompts have scaled models using autoregressive (Shen et al., 2024; Le et al., 2024; Ju et al., 2024; Lee et al., 2024; Yang et al., 2024; Eskimez et al., 2024; Liu et al., 2024) and diffusion frameworks (Jiang et al., 2023b; Wang et al., 2023a;c; Jiang et al., 2023a; Peng et al., 2024; Kim et al., 2024; Chen et al., 2024b; Meng et al., 2024; Yang et al., 2024; Lovelace et al., 2023; Liu et al., 2024). While these models scale well, they suffer from slow inference due to iterative sampling. Our model, DMDSpeech, addresses this by combining the scalability of prompt-based methods with the efficiency of non-iterative sampling. Through distribution matching distillation, we transform a diffusion-based TTS model into a student model capable of generating speech in a few steps, accelerating inference and enabling direct metric optimization for state-of-the-art speaker similarity and speech quality.

**Diffusion Distillation**    Diffusion models generate high-quality audio but suffer from slow inference due to iterative sampling (Popov et al., 2021). Diffusion distillation accelerates this process by training a student model to efficiently replicate the teacher's behavior. Previous methods approximated the teacher's ODE sampling trajectories. ProDiff (Huang et al., 2022) used progressive distillation (Salimans & Ho, 2022) to reduce sampling steps, while CoMoSpeech (Ye et al., 2023) and FlashSpeech (Ye et al., 2024) employed consistency distillation (Song et al., 2023). Rectified flow methods, such as in VoiceFlow (Guo et al., 2024) and ReFlow-TTS (Guan et al., 2024), aimed to accelerate sampling by straightening sampling paths. However, these methods often compromise quality by forcing the student to follow the teacher's path, which may not suit its reduced capacity. An alternative is distribution matching, either adversarially (Sauer et al., 2023; 2024) or via score function matching (Yin et al., 2024b), aligning the student with the teacher in distribution to maintain quality. However, these methods may reduce diversity as the student model might prioritize high-probability regions

of the distribution. Additionally, many distillation techniques require generating noise-data pairs (Sauer et al., 2024; Yin et al., 2024b; Liu et al., 2023), which is computationally expensive. We utilize DMD2 (Yin et al., 2024a), which bypasses the need for pair generation and enhances quality through adversarial training. Interestingly, we find that the reduction in diversity through DMD distillation can also reduce the chance of sampling from low-probability regions of the distributions, which can lead to unwanted artifacts and hallucinations. Consequently, distillation may enhance the human perceived quality, similar to recent findings in autoregressive diffusion models (Liu et al., 2024).

**Direct Metric Optimization**    Optimizing generative models using perceptual metrics has gained attention recently. MetricGAN (Fu et al., 2019) optimized speech enhancement models using PESQ and STOI as rewards in an adversarial setting. Reinforcement learning from human feedback (RLHF) has also been used to improve speech naturalness by optimizing predicted MOS scores (Zhang et al., 2024; Chen et al., 2024a). However, these approaches are challenging to apply to many state-of-the-art models due to non-differentiable components like duration upsamplers (Li et al., 2024b; Ye et al., 2024) or iterative sampling (Lee et al., 2024; Peng et al., 2024). In contrast, DMDSpeech allows end-to-end differentiable speech generation without iterative processes. We employ speaker verification (SV) loss and connectionist temporal classification (CTC) loss to optimize speaker similarity and text-speech alignment. By directly optimizing these metrics, we significantly improve speaker similarity and intelligibility, aligning synthesized speech more closely with human auditory preferences, marking the first application of direct metric optimization in speech synthesis models.

## 3 METHODS

### 3.1 PRELIMINARY: END-TO-END LATENT SPEECH DIFFUSION

Our model starts with a pre-trained teacher model based on an end-to-end latent speech diffusion framework such as SimpleTTS (Lovelace et al., 2023) and DiTTo-TTS (Lee et al., 2024). This section outlines the formulation of the diffusion process, noise scheduling, and the objective function.

We begin by encoding raw audio waveforms $\mathbf{y} \in \mathbb{R}^{1 \times T}$, where $T$ is the audio length, into latent representations $\mathbf{x}_0 = \mathcal{E}(\mathbf{y})$ using a latent autoencoder $\mathcal{E}$. The latent autoencoder follows DAC (Kumar et al., 2024) with residual vector quantization replaced by the variational autoencoder loss (see Appendix C.1 for more information). We denote the ground truth latent distribution as $p_{\text{data}}$. The diffusion process involves adding noise to $\mathbf{x}_0 \sim p_{\text{data}}$ over continuous time $t \in [0, 1]$ through a noise schedule. Our noise schedule follows Lovelace et al. (2023), which is a shifted cosine noise schedule formulated with $\alpha_t$ and $\sigma_t$ that control the amount of signal and noise (see Appendix C.2.1).

During training, the model learns to remove noise added to the latent representations. Given a latent variable $\mathbf{x}_0$ and noise $\boldsymbol{\epsilon} \sim \mathcal{N}(\mathbf{0}, \mathbf{I})$, the noisy latent $\mathbf{x}_t$ at time step $t$ is generated as $\mathbf{x}_t = \alpha_t \mathbf{x}_0 + \sigma_t \boldsymbol{\epsilon}$. We use a binaray prompt mask $\mathbf{m}$ to selectively preserve the original values in regions corresponding to the prompt. The noisy latent $\mathbf{x}_t$ is adjusted as $\mathbf{x}_t \leftarrow \mathbf{x}_t \odot (1 - \mathbf{m}) + \mathbf{x}_0 \odot \mathbf{m}$, where $\odot$ denotes element-wise multiplication. The binary mask $\mathbf{m}$ is randomly sampled to mask between 0% to 50% of the length of $\mathbf{x}_0$. We define a reparameterized velocity $\mathbf{v} = \alpha_t \boldsymbol{\epsilon} - \sigma_t \mathbf{x}_0$, which serves as the training objective as in Huang et al. (2022). We train our diffusion transformer (Peebles & Xie, 2023) model $f_\phi$, parameterized by $\phi$, to predict the target $\mathbf{v}$ given the noisy latent $\mathbf{x}_t$, conditioned on text embeddings $\mathbf{c}$, prompt mask indicators $\mathbf{m}$, and the time step $t$:

$$\mathcal{L}_{\text{diff}}(f_\phi; p_{\text{data}}) = \mathbb{E}_{\mathbf{x}_0 \sim p_{\text{data}}, t \sim \mathcal{U}(0,1), \boldsymbol{\epsilon} \sim \mathcal{N}(0,I)} \left[ \|\mathbf{v} - f_\phi(\mathbf{x}_t; \mathbf{c}, \mathbf{m}, t)\|_2 \right]. \quad (1)$$

During inference, the model takes noise $\mathbf{z} \in \mathcal{N}(0, I)$ with fixed size $[d, L]$ where $L$ is the total duration of the target speech. $L$ is estimated by multiplying the number of phonemes in the target text with the speaking rate of the prompt speech (see Appendix C.2.3 for more implementation details).

### 3.2 IMPROVED DISTRIBUTION MATCHING DISTILLATION

We employ improved Distribution Matching Distillation (Yin et al., 2024a), or DMD 2, to distill our teacher model for fast sampling and direct metric optimization. DMD 2 improves upon DMD (Yin et al., 2024b) by incorporating adversarial training on the real data, eliminating the need for noise-data pair generation and significantly reducing the training cost. This section details how we adapt DMD for efficient speech synthesis, including the formulations corresponding to our implementation.

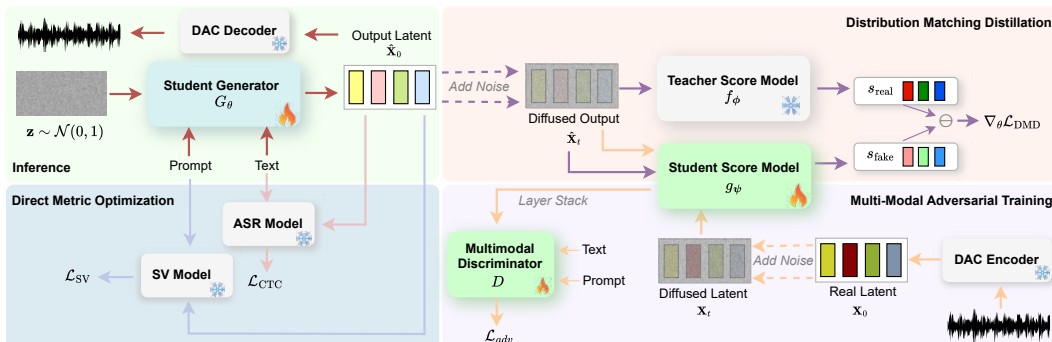

Figure 1: Overview of the DMDSpeech framework. The framework consists of inference and three main components for training: (1) **Inference** (upper left): A few-step distilled generator $G_\theta$ synthesizes speech directly from noise, conditioned on the text and speaker prompt (red arrow). (2) **Distribution Matching Distillation** (upper right): Gradient computation for DMD loss where the student score model $g_\psi$ matches the teacher score model $f_\phi$ to align the distribution of student generator $G_\theta$ with the teacher distribution (purple arrow). (3) **Multi-Modal Adversarial Training** (lower right): The discriminator $D$ uses stacked features from the student score model to distinguish between real and synthesized noisy latents conditioned on both text and prompt (yellow arrows). (4) **Direct Metric Optimization** (lower left): Direct metric optimization for word error rate (WER) via CTC loss (pink arrow) and speaker embedding cosine similarity (SIM) via SV loss (blue arrow).

**Background on Distribution Matching Distillation**    DMD aims to train a student generator $G_\theta$ to produce samples whose distribution matches the data distribution $p_{\text{data}}$ after a forward diffusion process. The objective is to minimize the Kullback-Liebler (KL) divergence between the distributions of the diffused real data $p_{\text{data},t}$ and the diffused student generator outputs $p_{\theta,t}$ across all time $t \in [0,1]$:

$$\mathcal{L}_{\text{DMD}} = \mathbb{E}_{t\sim\mathcal{U}(0,1)}\left[D_{KL}(p_{\theta,t}||p_{\text{data},t})\right] = \mathbb{E}_{t\sim\mathcal{U}(0,1)}\left[\mathbb{E}_{\mathbf{x}\sim p_{\theta,t}}\left[\log\left(\frac{p_{\theta,t}(\mathbf{x})}{p_{\text{data},t}(\mathbf{x})}\right)\right]\right] \quad (2)$$

$$= -\mathbb{E}_{t\sim\mathcal{U}(0,1)}\left[\mathbb{E}_{\mathbf{x}\sim p_{\theta,t}}\left[\log\left(p_{\text{data},t}(\mathbf{x})\right) - \log\left(p_{\theta,t}(\mathbf{x})\right)\right]\right]. \quad (3)$$

Since DMD trains $G_\theta$ through gradient descent, the formulation DMD only requires the gradient of the DMD loss with respect to the generator parameters $\theta$, which is derived in Yin et al. (2024b) as:

$$\nabla_\theta \mathcal{L}_{\text{DMD}} = -\mathbb{E}_{t\sim\mathcal{U}(0,1)}\left[\mathbb{E}_{\mathbf{x}_t\sim p_{\theta,t},\mathbf{z}\sim\mathcal{N}(\mathbf{0},\mathbf{I})}\left[\omega_t\alpha_t\left(s_{\text{real}}(\mathbf{x}_t,t) - s_\theta(\mathbf{x}_t,t)\right)\nabla_\theta G_\theta(\mathbf{z})\right]\right], \quad (4)$$

where $\mathbf{x}_t$ is the diffused version of $\mathbf{x}_0 = G_\theta(\mathbf{z})$, the distilled generator output for $\mathbf{z}\sim\mathcal{N}(\mathbf{0},\mathbf{I})$, $s_{\text{real}}(\mathbf{x}_t,t)$ and $s_\theta(\mathbf{x}_t,t)$ are neural network approximation of score functions of the diffused data distribution and student output distribution, and $\omega_t$ is a weighting factor defined in eq. 32.

In our speech synthesis task, the generator $G_\theta$ produces latent speech representations $\mathbf{x}_0$ conditioned on input text $\mathbf{c}$ and a speaker prompt. The teacher diffusion model $f_\phi$ serves as the score function $s_{\text{real}}$ for the real data distribution. We train another diffusion model $g_\psi$ to approximate the score of the distilled generator's output distribution $p_\theta$ following eq. 1. The scores are estimated as:

$$s_{\text{real}}(\mathbf{x}_t,t) = -\frac{\mathbf{x}_t - \alpha_t\hat{\mathbf{x}}_0^{\text{real}}}{\sigma_t^2}, \quad (5) \qquad s_\theta(\mathbf{x}_t,t) = -\frac{\mathbf{x}_t - \alpha_t\hat{\mathbf{x}}_0^{\text{fake}}}{\sigma_t^2}. \quad (6)$$

where $\hat{\mathbf{x}}_0^{\text{real}}$ and $\hat{\mathbf{x}}_0^{\text{fake}}$ are estimation of $\mathbf{x}_0$ from the teacher and student diffusion models, respectively:

$$\hat{\mathbf{x}}_0^{\text{real}} = \frac{\mathbf{x}_t - \sigma_t f_\phi(\mathbf{x}_t\,;\mathbf{c},\mathbf{m},t)}{\alpha_t}, \quad (7) \qquad \hat{\mathbf{x}}_0^{\text{fake}} = \frac{\mathbf{x}_t - \sigma_t g_\psi(\mathbf{x}_t\,;\mathbf{c},\mathbf{m},t)}{\alpha_t}. \quad (8)$$

The parameters of $G_\theta$ and $g_\psi$ are both initialized from the teacher diffusion model's parameters $\phi$.

**DMD 2 for Speech Synthesis**    We notice that the one-step student model results in noticeable artifacts, as the student model lacks the computational capacity to capture all the acoustic details that the teacher model generates through multiple iterative steps. To address this issue, we adopt the DMD 2 framework from Yin et al. (2024a) by conditioning the student generator $G_\theta$ on the noise level $t$. This conditioning allows the model to estimate the clean latent speech representation $\mathbf{x}_0$ from

its noisy counterpart $\mathbf{x}_t$ for a sequence of predefined time steps $t \in \{t_1, \ldots, t_N\}$. This multi-step sampling process (Algorithm 1) is similar to the consistency model proposed by Song et al. (2023).

The process goes as follows: for each time step $t_n$, the student model produces an estimate $\hat{\mathbf{x}}_0^n = G_\theta(\mathbf{x}_{t_n} ; \mathbf{c}, \mathbf{m}, t_n)$, and this estimate is then re-noised to obtain $\mathbf{x}_{t_{n+1}}$ as input for the next time step:

$$\mathbf{x}_{t_{n+1}} = \alpha_{t_{n+1}} \hat{\mathbf{x}}_0^n + \sigma_{t_{n+1}} \boldsymbol{\epsilon}, \quad \boldsymbol{\epsilon} \sim \mathcal{N}(0, \mathbf{I}). \tag{9}$$

This process generates progressively less noisy versions of $\mathbf{x}_0$ at decreasing noise levels $\sigma_{t_{n+1}} < \sigma_{t_n}$.

For our four-step model, we use a schedule of time steps $\{1.0, 0.75, 0.50, 0.25\}$ mapped from the teacher's full range $t \in [0, 1]$. We simulate the one-step inference during training to minimize the training/inference mismatch. Instead of using the noisy version of ground truth $\alpha_{t_n} \mathbf{x}_0 + \sigma_{t_n} \boldsymbol{\epsilon}$ as input, we use the noisy version of student prediction $\alpha_{t_n} G_\theta \left( \alpha_{t_{n-1}} \mathbf{x}_0 + \sigma_{t_{n-1}} \boldsymbol{\epsilon} ; \mathbf{c}, \mathbf{m}, t_{n-1} \right) + \sigma_{t_n} \boldsymbol{\epsilon}$ from the noisy ground truth at the noise level $\sigma_{n-1} > \sigma_n$. This is different from Yin et al. (2024a), which simulates all four steps, as we found that simulating just one step is sufficient for producing high-quality speech while saving GPU memory during training.

To further improve the performance of the student model, we incorporate adversarial training following the approach of Yin et al. (2024a) that allows the students to learn from the real data. However, unlike in text-to-image synthesis, where text acts as a weak condition for the generated image, text-to-speech synthesis requires strong conditioning on both text and speaker prompt. The generated speech must strictly adhere to the semantic content of the text and the prompt speaker's voice and style. To this end, we modify the adversarial discriminator used in Yin et al. (2024a) to a conditional multimodal discriminator, inspired by Janiczek et al. (2024). Following Li et al. (2024b), our discriminator $D$ is a conformer that takes as input the stacked features from all transformer layers of the student score network $g_\psi$ with noisy input, along with the text embeddings $\mathbf{c}$, prompt mask $\mathbf{m}$, and noise level $t$ (denoted as $\mathcal{C}$). The discriminator is trained with the LSGAN loss (Mao et al., 2017):

$$\mathcal{L}_{\mathrm{adv}}(G_\theta; D) = \mathbb{E}_t \left[ \mathbb{E}_{\hat{\mathbf{x}}_t \sim p_{\theta, t,}, \mathbf{m}} \left[ \left( D \left( \tilde{g}_\psi(\hat{\mathbf{x}}_t ; \mathcal{C}) ; \mathcal{C} \right) - 1 \right)^2 \right] \right], \tag{10}$$

$$\mathcal{L}_{\mathrm{adv}}(D; G_\theta) = \mathbb{E}_t \left[ \mathbb{E}_{\hat{\mathbf{x}}_t \sim p_{\theta, t}, \mathbf{m}} \left[ \left( D \left( \tilde{g}_\psi(\hat{\mathbf{x}}_t ; \mathcal{C}) ; \mathcal{C} \right) \right)^2 \right] \right] + \tag{11}$$

$$\mathbb{E}_t \left[ \mathbb{E}_{\mathbf{x}_t \sim p_{\mathrm{data}, t}, \mathbf{m}} \left[ \left( D \left( \tilde{g}_\psi(\mathbf{x}_t ; \mathcal{C}) ; \mathcal{C} \right) - 1 \right)^2 \right] \right], \tag{12}$$

where $\mathcal{C} = \{\mathbf{c}, \mathbf{m}, t\}$ is the conditional input, $\tilde{g}_\psi(\cdot)$ denotes the stacked features from all layers of $g_\psi$, and $\hat{\mathbf{x}}_t = \alpha_t G_\theta(\mathbf{z}; \mathcal{C}) + \sigma_t \boldsymbol{\epsilon}$ is the noisy version of the student-generated speech at time step $t$.

### 3.3 Direct Metric Optimization

We directly optimize two metrics, speaker embedding cosine similarity (SIM) and word error rate (WER), which are commonly used for evaluating zero-shot speech synthesis models and are both shown to correlate with human perception for speaker similarity (Thoidis et al., 2023) and naturalness (Alharthi et al., 2023). To improve WER, we incorporate a Connectionist Temporal Classification (CTC) loss (Graves et al., 2006). The CTC loss aligns the synthesized speech with the input text at the character level, reducing word error rates and enhancing robustness. It is defined as:

$$\mathcal{L}_{\mathrm{CTC}} = \mathbb{E}_{\mathbf{x}_{\mathrm{fake}} \sim p_\theta, \mathbf{c}} \left[ -\log p(\mathbf{c}|C(\mathbf{x}_{\mathrm{fake}})) \right], \tag{13}$$

where $\mathbf{x}_{\mathrm{fake}}$ is the student-generated speech, $\mathbf{c}$ is the text transcript, and $C(\cdot)$ is a pre-trained CTC-based ASR model on speech latent (see Appendix C.3 for details). We also employ a Speaker Verification (SV) loss to ensure the synthesized speech matches the target speaker's identity. We use a pre-trained speaker verification model $S$ on latent (see Appendix C.4 for details) for the SV loss:

$$\mathcal{L}_{\mathrm{SV}} = \mathbb{E}_{\substack{\mathbf{x}_{\mathrm{real}} \sim p_{\mathrm{data}}, \\ \mathbf{x}_{\mathrm{fake}} \sim p_\theta, \mathbf{m}}} \left[ 1 - \frac{\mathbf{e}_{\mathrm{real}} \cdot \mathbf{e}_{\mathrm{fake}}}{\|\mathbf{e}_{\mathrm{real}}\| \, \|\mathbf{e}_{\mathrm{fake}}\|} \right], \quad \mathbf{e}_{\mathrm{fake}} = S(\mathbf{x}_{\mathrm{fake}}), \quad \mathbf{e}_{\mathrm{real}} = S(\mathbf{x}_{\mathrm{real}} \odot \mathbf{m}), \tag{14}$$

where $\mathbf{e}_{\mathrm{real}}$ and $\mathbf{e}_{\mathrm{fake}}$ are the speaker embeddings of the prompt and student-generated speech.

### 3.4 Training Objectives and Stability

The overall training objective for $G_\theta$ combines DMD and adversarial losses with SV and CTC losses:

$$\min_\theta \ \mathcal{L}_{\mathrm{DMD}} + \lambda_{\mathrm{adv}} \mathcal{L}_{\mathrm{adv}}(G_\theta; D) + \lambda_{\mathrm{SV}} \mathcal{L}_{\mathrm{SV}} + \lambda_{\mathrm{CTC}} \mathcal{L}_{\mathrm{CTC}}, \tag{15}$$

and the training objectives for $g_\psi$ and $D$ are:

$$\min_{\psi}\ \mathcal{L}_{\text{diff}}\left(g_\psi; p_\theta\right), \qquad (16) \qquad\qquad \min_{D}\ \mathcal{L}_{\text{adv}}\left(D; G_\theta\right). \qquad (17)$$

We employ an alternating training strategy where the student generator $G_\theta$, the student score estimator $g_\psi$, and the discriminator $D$ are updated at different frequencies to maintain training stability. Specifically, for every update of $G_\theta$, we perform five updates of $g_\psi$. This ensures that the score estimator $g_\psi$ can adapt quickly to the dynamic changes in the generator distribution $p_\theta$. Unlike Yin et al. (2024a), where $D$ are updated five times for every single update of $G_\theta$, we update $D$ and $G_\theta$ at the same rate. This prevents the discriminator from becoming too powerful and destabilizing training.

The learning rates for $G_\theta$ and $g_\psi$ play a critical role in maintaining training stability since both models are initialized from the teacher's parameters, $\phi$. Treating this as a fine-tuning process, we set their learning rates close to the teacher model's final learning rate to prevent catastrophic forgetting and training collapse. The teacher model was trained using a cosine annealing warmup scheduler, which gradually reduced the learning rate over time. Thus, starting with a high learning rate for $G_\theta$ and $g_\psi$ can cause them to deviate significantly from the pre-trained knowledge, leading to training failure. Conversely, the learning rate for $D$ is less sensitive and does not require such precise tuning.

Balancing the different loss components in the overall objective function is crucial for successful training. The primary loss, $\mathcal{L}_{\text{DMD}}$, is responsible for transferring knowledge from the teacher model, aligning the synthesized speech with the text. Other losses, such as $\mathcal{L}_{\text{adv}}$, $\mathcal{L}_{\text{SV}}$, and $\mathcal{L}_{\text{CTC}}$, need to be scaled properly to match the gradient of $\mathcal{L}_{\text{DMD}}$. We set $\lambda_{\text{adv}} = 10^{-3}$ to ensure the gradient norm of $\mathcal{L}_{\text{adv}}$ is comparable to that of $\mathcal{L}_{\text{DMD}}$. During early training stage, we observed that the gradient norms of $\mathcal{L}_{\text{SV}}$ and $\mathcal{L}_{\text{CTC}}$ were significantly higher than $\mathcal{L}_{\text{DMD}}$, likely because $G_\theta$ was still learning to generate intelligible speech from single step. To address this, we set $\lambda_{\text{CTC}} = 0$ for the first 5,000 iterations and $\lambda_{\text{SV}} = 0$ for the first 10,000 iterations. This allows $G_\theta$ to stabilize under the influence of $\mathcal{L}_{\text{DMD}}$ before integrating these additional losses. After that, both $\lambda_{\text{CTC}}$ and $\lambda_{\text{SV}}$ are set to 1.

## 4 EXPERIMENTS

### 4.1 MODEL TRAINING

We conducted our experiments on the LibriLight dataset (Kahn et al., 2020), which consists of 57,706.4 hours of audio from 7,439 speakers. The data and transcripts were obtained using Python scripts provided by the LibriLight authors[1]. All audio files were resampled to 48 kHz to match the configuration of our DAC autoencoder, and the text was converted into phonemes using Phonemizer (Bernard & Titeux, 2021). To manage memory constraints, we segmented the audio into 30-second chunks using WhisperX (Bain et al., 2023). The teacher model $f_\phi$ was trained for 400,000 steps with a batch size of 384, using the AdamW optimizer (Loshchilov & Hutter, 2018) with $\beta_1 = 0.9$, $\beta_2 = 0.999$, weight decay of $10^{-2}$, and an initial learning rate of $10^{-4}$. The learning rate followed a cosine decay schedule with a 4,000-step warmup, gradually decreasing to $10^{-5}$. Model weights were updated using an exponential moving average (EMA) with a decay factor of 0.99 every 100 steps. The teacher model consists of 450M parameters in total. For student training, we initialized both the student generator $G_\theta$ and the student score model $g_\psi$ with the EMA-weighted teacher parameters. The initial learning rate was set to match the final learning rate of the teacher model ($\lambda = 10^{-5}$), while the batch size was reduced to 96 due to memory constraints. Reducing the batch size further negatively impacted performance, as a sufficiently large batch size is required for accurate score estimation due to the Monte Carlo nature of DMD (see Section 4.4 for further discussion). The student generator $G_\theta$ and the discriminator $D$ were trained for an additional 40,000 steps, and the student score model $g_\psi$ for 200,000 steps accordingly using the same optimization settings as the teacher. All models were trained on 24 NVIDIA A100 40GB GPUs.

### 4.2 EVALUATION METRICS

We performed both subjective and objective evaluations to assess the performance of our model and several state-of-the-art baselines. For subjective evaluation, we employed four metrics rated on a

---

[1]Available at `https://github.com/facebookresearch/libri-light/`

Table 1: Comparison between our models and non-E2E baselines on four subjective metrics: naturalness (MOS-N), sound quality (MOS-Q), voice similarity (SMOS-V), and speaking style similarity (SMOS-S). Scoes are presented as means ($\pm$ standard error). One asterisk (*) indicates a statistically significant difference ($p < 0.05$) and double asterisk (**) indicates $p < 0.01$ compared to DMDSpeech. The best models and those within one standard error of the best are highlighted.

| Model | MOS-N | MOS-Q | SMOS-V | SMOS-S |
|---|---|---|---|---|
| Ground Truth | 4.47 ($\pm$ 0.03) | 4.61 ($\pm$ 0.03) | 3.86 ($\pm$ 0.05)** | 3.81 ($\pm$ 0.05)** |
| Ours (DMDSpeech, N=4) | **4.42 ($\pm$ 0.03)** | **4.59 ($\pm$ 0.03)** | **4.49 ($\pm$ 0.03)** | **4.30 ($\pm$ 0.03)** |
| Ours (Teacher, N=128) | 4.32 ($\pm$ 0.04)* | 4.55 ($\pm$ 0.03) | 4.17 ($\pm$ 0.04)** | 4.00 ($\pm$ 0.04)** |
| NaturalSpeech 3 (Ju et al., 2024) | 4.24 ($\pm$ 0.04)** | 4.55 ($\pm$ 0.03) | 4.44 ($\pm$ 0.03) | 4.25 ($\pm$ 0.04) |
| StyleTTS-ZS (Li et al., 2024b) | **4.40 ($\pm$ 0.03)** | 4.54 ($\pm$ 0.03) | 4.34 ($\pm$ 0.04)** | 4.20 ($\pm$ 0.03)* |

Table 2: Comparison between our models and end-to-end baseline models.

| Model | MOS-N | MOS-Q | SMOS-V | SMOS-S |
|---|---|---|---|---|
| Ground Truth | 4.37 ($\pm$ 0.03)* | 4.49 ($\pm$ 0.03) | 3.51 ($\pm$ 0.05)** | 3.39 ($\pm$ 0.05)** |
| Ours (DMDSpeech, N=4) | **4.27 ($\pm$ 0.03)** | **4.45 ($\pm$ 0.03)** | **4.35 ($\pm$ 0.03)** | **4.16 ($\pm$ 0.03)** |
| Ours (Teacher, N=128) | 4.22 ($\pm$ 0.04) | 4.40 ($\pm$ 0.03) | 4.03 ($\pm$ 0.04)** | 3.87 ($\pm$ 0.04)** |
| DiTTo-TTS (Lee et al., 2024) | **4.28 ($\pm$ 0.04)** | 4.41 ($\pm$ 0.03) | 4.16 ($\pm$ 0.04)** | 4.07 ($\pm$ 0.03)* |
| VoiceCraft (Peng et al., 2024) | 3.76 ($\pm$ 0.05)** | 3.88 ($\pm$ 0.04)** | 3.41 ($\pm$ 0.05)** | 3.37 ($\pm$ 0.05)** |
| CLaM-TTS (Kim et al., 2024) | 3.77 ($\pm$ 0.05)** | 3.87 ($\pm$ 0.04)** | 3.67 ($\pm$ 0.05)** | 3.43 ($\pm$ 0.05)** |
| XTTS (Casanova et al., 2024) | 3.63 ($\pm$ 0.05)** | 3.89 ($\pm$ 0.04)** | 3.25 ($\pm$ 0.05)** | 3.22 ($\pm$ 0.05)** |

scale from 1 to 5. The Mean Opinion Score for Naturalness (MOS-N) assessed the human-likeness of the synthesized speech, where 1 indicates fully synthesized audio and 5 indicates completely human speech. The Mean Opinion Score for Sound Quality (MOS-Q) evaluated audio quality degradation relative to the prompt, with 1 representing severe degradation and 5 indicating no degradation. The Similarity Mean Opinion Score for Voice (SMOS-V) measured the similarity of the synthesized voice to the prompt speaker's voice, where 1 means completely different and 5 means identical. Lastly, the Similarity Mean Opinion Score for Style (SMOS-S) assessed the speaking style similarity to the prompt speaker with the same scale. These subjective evaluations were conducted through a listening test survey on the crowdsourcing platform Prolific, with 1,000 tests (30 samples each) taken by native English speakers with no hearing impairments who had experience in content creation or audio/video editing, ensuring they could better differentiate synthesized audio from real human. The prompt speech served as an anchor that is supposed to score 5 on all metrics; we also included intentionally mismatched speakers serving as low anchor for similarity, which should have a rating lower than 3. The participants who fails to correctly rate the anchors hidden in the test are disqualified and their answers removed (details in Appendix E.2). For objective evaluation, we followed the approach from previous works (Wang et al., 2023a; Lee et al., 2024) and measured speaker similarity using the cosine similarity between speaker embeddings of the generated speech and the promot (SIM), using the WavLM-TDCNN speaker embedding model[2]. We also calculated the Word Error Rate (WER) with a CTC-based HuBERT ASR model [3] following (Ju et al., 2024; Shen et al., 2024).

### 4.3 COMPARISON TO OTHER MODELS

We conducted two evaluation experiments to compare our models against two categories of baselines: state-of-the-art (SOTA) non-end-to-end (E2E) models that include explicit duration and prosody modeling, and recent E2E models without such explicit modeling. For both experiments, the samples were downsampled to 16 kHz for fairness and prompts were transcribed using WhisperX for synthesis.

In the first experiment, we compared our model to NaturalSpeech 3 and StyleTTS-ZS, both of which utilize explicit duration and prosody modeling and were trained on the large-scale LibriLight dataset.

---

[2]WavLM large fine-tuned checkpoint: `https://github.com/microsoft/UniSpeech/tree/main/downstreams/speaker_verification`

[3]`https://huggingface.co/facebook/hubert-large-ls960-ft`

Table 3: Objective evaluation results between our models and other baseline models. For training data, LL stands for LibriLight, G stands for GigaSpeech, and assorted means combination of various datasets (see Appendix E.1 for details). The real-time factor (RTF) was computed on a NVIDIA V100 GPU except DiTTo-TTS and CLaM-TTS, whose RTF is obtained from their papers using the inference time needed to synthesize 10s of speech divided by 10 on unknown devices. Additional evaluation results on emotion reflection are presented in Table 6.

| Model | Training Set | # Parameters | WER ↓ | SIM ↑ | RTF ↓ |
|---|---|---|---|---|---|
| Ground Truth | — | — | 2.19 | 0.67 | — |
| Ours (DMDSpeech, N=4) | LL (∼58k hrs) | 450M | 1.94 | **0.69** | **0.07** |
| Ours (Teacher, N=128) | LL (∼58k hrs) | 450M | 9.51 | 0.55 | 0.96 |
| NaturalSpeech 3 | LL (∼58k hrs) | 500M | **1.81** | 0.67 | 0.30 |
| VoiceCraft | G+LL (∼69k hrs) | 830M | 6.32 | 0.61 | 1.12 |
| DiTTo-TTS | Assorted (∼56k hrs) | 740M | 2.56 | 0.62 | 0.16 |
| CLaM-TTS | Assorted (∼56k hrs) | 584M | 5.11 | 0.49 | 0.42 |
| XTTS | Assorted (∼17k hrs) | 482M | 4.93 | 0.49 | 0.37 |

Since neither model has public source code or official checkpoints available, we used 47 official samples from the authors and other sources (details in Appendix E.1) from the LibriSpeech *test-clean* subset, covering all 40 speakers. As shown in Table 1, our distilled model significantly outperformed NaturalSpeech 3 in naturalness and StyleTTS-ZS in similarity metrics. It also outperformed the teacher model in terms of naturalness, voice similarity, and style similarity.

In the second experiment, we evaluated E2E speech synthesis models without explicit duration modeling, including three popular autoregressive models, XTTS , CLaM-TTS, VoiceCraft, and one diffusion-based SOTA model, DiTTo-TTS. Since official code and checkpoints for CLaM-TTS and DiTTo-TTS were unavailable, we obtained 3,711 samples from the authors based on the LibriSpeech *test-clean* subset[4] and synthesized the corresponding samples using XTTS, VoiceCraft, and our models. For subjective evaluation, we selected 80 samples, ensuring that each speaker from the *test-clean* subset was represented by two samples. As shown in Table 2, our model significantly outperformed all recent E2E speech synthesis baselines except DiTTo-TTS in MOS, with which it achieved comparable performance in naturalness and sound quality. This indicates that our model is consistently preferred across both naturalness and similarity by human listeners.

All baselines, except for NaturalSpeech 3, were evaluated using the 3,711 samples as per Lee et al. (2024). Since we lacked sufficient samples for a direct evaluation of NaturalSpeech 3, its results are taken from their original paper. Table 5 shows that our model achieved the highest speaker similarity score (SIM) to the prompt, even surpassing the ground truth. The Real-Time Factor (RTF) of the distilled model is 13.7 times lower than the teacher model, which is lower than all baseline methods by a large margin. Although our model had a slightly higher WER (1.94) compared to NaturalSpeech 3 (1.81), it is important to note that our model is entirely end-to-end without explicit duration modeling, unlike NaturalSpeech 3. Both DMDSpeech and NaturalSpeech 3 also exhibited lower WER than the ground truth. One point to consider is the high WER of our teacher model, which is mainly due to cutoff at the end of sentences in the training set caused by faulty segmentation with WhisperX. It affects about 10% of the utterances. After distillation, this issue was resolved due to mode shrinkage (discussed in Section 4.4). Moreover, our model demonstrates significantly faster inference speed compared to all baseline models, as it only requires four sampling steps.

## 4.4 ABLATION STUDY

We conducted ablation studies to assess the contribution of each proposed component, with results summarized in Table 4. We evaluated models trained solely with DMD 2 using one sampling step (DMD 2 only, N=1) and four sampling steps (DMD 2 only, N=4), as well as models trained with only CTC loss or SV loss on top of four-step DMD 2 model. Additionally, we examined the impact of reducing the batch size from 96 to 16 (B. S. 96 → 16). The ablation study used the same 80 samples for subjective evaluation as in the second experiment and 3,711 samples for objective evaluation. To

---

[4]Prompts and samples were generated according to instructions provided in `https://github.com/keonlee9420/evaluate-zero-shot-tts`

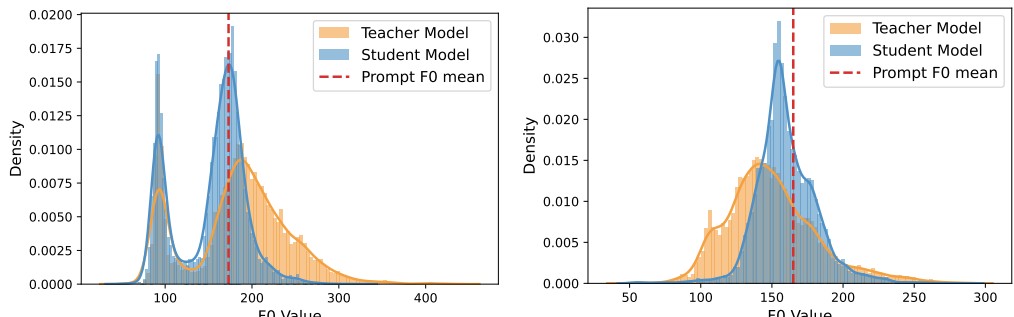

Figure 2: Illustration of mode shrinkage in terms of pitch. Speech with the same text and prompt were synthesized 50 times, and their frame-level F0 values are shown as histograms and kernel density estimates. The red dashed line represents the mean F0 value of the prompt. In both examples, the student's distribution shifts toward the most likely region, centering around the prompt's mean value.

measure the trade-off between speech diversity and model capacity, we included the coefficient of variation of pitch ($CV_{f_0}$). This metric was calculated by synthesizing speech with the same text and prompt 50 times and computing the coefficient of variation of the frame-level F0 values averaged over the speech frames. The final results reported were averaged over 40 prompts from the LibriSpeech *test-clean* subset, covering all 40 speakers.

**Effects of Distribution Matching Distillation**    Using a single sampling step resulted in significantly degraded performance compared to the full DMDSpeech model. While increasing to four sampling steps improved naturalness and sound quality to approach the teacher model's level, speaker similarity remained significantly lower. Interestingly, the speaker verification model's SIM score showed only a slight decrease, suggesting a phenomenon we term *mode shrinkage* (Figure 2), where distillation emphasizes high-probability regions of the data distribution. This focus can result in a more generic speaker profile, reducing perceived uniqueness in the prompt speaker's voice, while maintaining global speaker features as reflected in the SIM score. To address this, we introduced speaker verification loss in this work to better capture the distinct characteristics of the prompt speaker.

Mode shrinkage also led to reduced diversity, as indicated by a lower $CV_{f_0}$ across student models compared to the teacher. There is also a trade-off between diversity and sample quality, as one-step student obtained close-to-teacher diversity despite its lowest sample quality. However, as shown in Figure 5, this reduction in diversity applies only when synthesizing speech from the same prompt and text. Given that zero-shot TTS is highly conditional, requiring strict adherence to the input text and speaker prompt, this reduction in diversity is not necessarily undesirable. As we found out in the subjective test, MOS-N increases even when diversity decreases. The distilled model achieves sufficient mode coverage across varying prompts and texts while benefiting from direct metric optimization and faster inference. Notably, mode shrinkage also corrected a cut-off issue in the teacher model, which mimicked the cutoff patterns

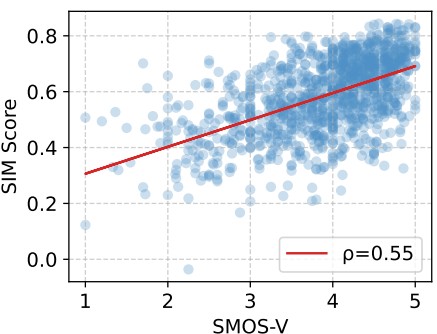

Figure 3: Scatter plot of human-rated voice similarity (SMOS-V) versus speaker embedding cosine similarity (SIM) at the utterance level. The correlation coefficient is 0.55.

in the training data. Since these cutoff samples represent a small portion of the dataset, they were significantly reduced by the student models during distillation, leading to a much lower word error rate. This observation prompted us to include CTC loss, further enhancing the model's intelligibility and robustness. For more discussion on mode shrinkage and its implications, see Appendix A.

Lastly, since DMD training involves estimating the score functions from training data through Monte Carlo simulation in a mini-batch, the batch size plays a critical role in the accuracy of distribution matching. Reducing the batch size from 96 to 16 significantly decreases sound quality and speaker similarity. Maintaining a sufficiently large batch size is crucial for stable DMD training.

Table 4: Ablation study comparing our proposed model with different conditions. MOS-N, MOS-Q, SMOS-V, and SMOS-S are reported as mean ($\pm$ standard error). Models with statistically significant differences ($p < 0.05$) compared to DMDSpeech are marked with one asterisk (*). Additional evaluation results on emotion reflection are presented in Table 7.

| Model | MOS-N | MOS-Q | SMOS-V | SMOS-S | WER | SIM | $CV_{f_0}$ |
|---|---|---|---|---|---|---|---|
| Teacher (N=128) | 4.22 ($\pm$ 0.04) | 4.40 ($\pm$ 0.03) | 4.03 ($\pm$ 0.04)* | 3.87 ($\pm$ 0.04)* | 9.51 | 0.55 | **0.70** |
| DMD 2 only (N=1) | 3.11 ($\pm$ 0.05)* | 2.99 ($\pm$ 0.05)* | 2.57 ($\pm$ 0.05)* | 2.74 ($\pm$ 0.05)* | 5.93 | 0.42 | 0.68 |
| DMD 2 only (N=4) | 4.19 ($\pm$ 0.03)* | 4.43 ($\pm$ 0.04) | 3.69 ($\pm$ 0.05)* | 3.62 ($\pm$ 0.05)* | 5.67 | 0.53 | 0.61 |
| +$\mathcal{L}_{CTC}$ only | 4.25 ($\pm$ 0.04) | 4.42 ($\pm$ 0.03) | 3.73 ($\pm$ 0.05)* | 3.62 ($\pm$ 0.05)* | **1.79** | 0.55 | 0.57 |
| +$\mathcal{L}_{SV}$ only | 4.07 ($\pm$ 0.04)* | 4.33 ($\pm$ 0.03) * | **4.35** ($\pm$ 0.04) | 4.15 ($\pm$ 0.04) | 6.62 | **0.70** | 0.61 |
| DMDSpeech (N=4) | **4.27** ($\pm$ **0.03**) | **4.45** ($\pm$ **0.03**) | **4.35** ($\pm$ **0.03**) | **4.16** ($\pm$ **0.03**) | 1.94 | 0.69 | 0.58 |
| B. S. 96 $\rightarrow$ 16 | 4.20 ($\pm$ 0.04) | 4.30 ($\pm$ 0.03)* | 4.27 ($\pm$ 0.04)* | 4.11 ($\pm$ 0.04) | 3.38 | 0.67 | 0.60 |

**Effects of Direct Metric Optimization**    We first demonstrate that the metrics we directly optimize are significantly correlated with human subjective ratings at the utterance level. Figure 3 shows the scatter plot between human-rated similarity SMOS-V and SIM, one of the optimized metrics, with a correlation coefficient $\rho = 0.55$. Another metric, word error rate (WER), is significantly correlated with naturalness (MOS-N) even at the utterance level, with a correlation $\rho = -0.15$ (see Figure 5). These correlations suggest a notable impact of these metrics on their associated subjective ratings.

When using only the CTC loss, we observe a substantial reduction in WER (from 5.67 to 1.79), but no improvement in speaker similarity, alongside a slight reduction in diversity and a minor improvement in naturalness. This aligns with the correlation between WER and human-rated naturalness with $\rho = -0.15$ ($p \ll 0.01$). In contrast, with only the SV loss, we see significant improvements in all speaker similarity metrics (SMOS-V, SMOS-S, SIM), but these gains come with a decrease in naturalness and sound quality, as well as an increase in WER. This suggests that while SV loss can enhance speaker similarity, it negatively impacts intelligibility and naturalness. Therefore, combining both CTC and SV losses achieves a balance between these metrics, yielding the best overall performance, with improvements across speaker similarity, intelligibility, and naturalness.

## 5 CONCLUSIONS

In this work, we presented DMDSpeech, a distilled diffusion-based text-to-speech (TTS) model based on prompt continuation. By employing distribution matching distillation (DMD), our model generates high-quality speech in just 4 steps and enables direct metric optimization. Through speaker verification (SV) and connectionist temporal classification (CTC) losses, DMDSpeech significantly improves speaker similarity and text-speech alignment, outperforming several state-of-the-art baselines.

The ability to directly optimize any differentiable metrics offers substantial progress in bridging the gap between generative modeling and human perception. As these metrics continue to improve, the alignment with human auditory preferences is expected to strengthen. This creates promising future directions, such as using reinforcement learning from human feedback to further improve TTS systems. Additionally, developing new differentiable metrics that better capture human perception could provide more robust optimization targets, aligning models more closely with human preferences.

However, DMDSpeech raises important ethical concerns. Our model has demonstrated the ability to generate speech with higher perceived similarity to the prompt than real utterances by the same speaker, as judged by both human listeners and speaker verification systems. This highlights limitations in current speaker verification models and presents risks of misuse, such as deepfake generation. To mitigate these risks, more advanced speaker verification techniques capable of distinguishing synthetic from real speech are necessary, alongside robust watermarking to identify synthesized audio. Clear ethical guidelines and legal frameworks will also be crucial to prevent abuse in sensitive areas.

We also observed that while DMDSpeech benefits from fast sampling and direct metric optimization, this comes with a trade-off in speech diversity. The reduction in diversity, which arises from prioritizing sampling speed and quality, warrants further investigation and improvement. Scaling the model with larger datasets and incorporating diverse languages may help mitigate this trade-off.

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

# A   MODE SHRINKAGE

To further explore the effects of mode shrinkage, we conducted experiments on *unconditional* diversity and mode coverage. Specifically, we used a continuation task where the model was asked to generate speech following a truncated prompt with its full text transcription, allowing us to compare the generated speech to its corresponding ground truth from real speakers. We evaluated two key aspects of speech: pitch (F0) and energy. As shown in Figure 5, the student model closely matches the teacher's distribution in both F0 and energy, demonstrating minimal mode shrinkage in contrast to the results shown in Figure 2, where mode shrinkage was evident.

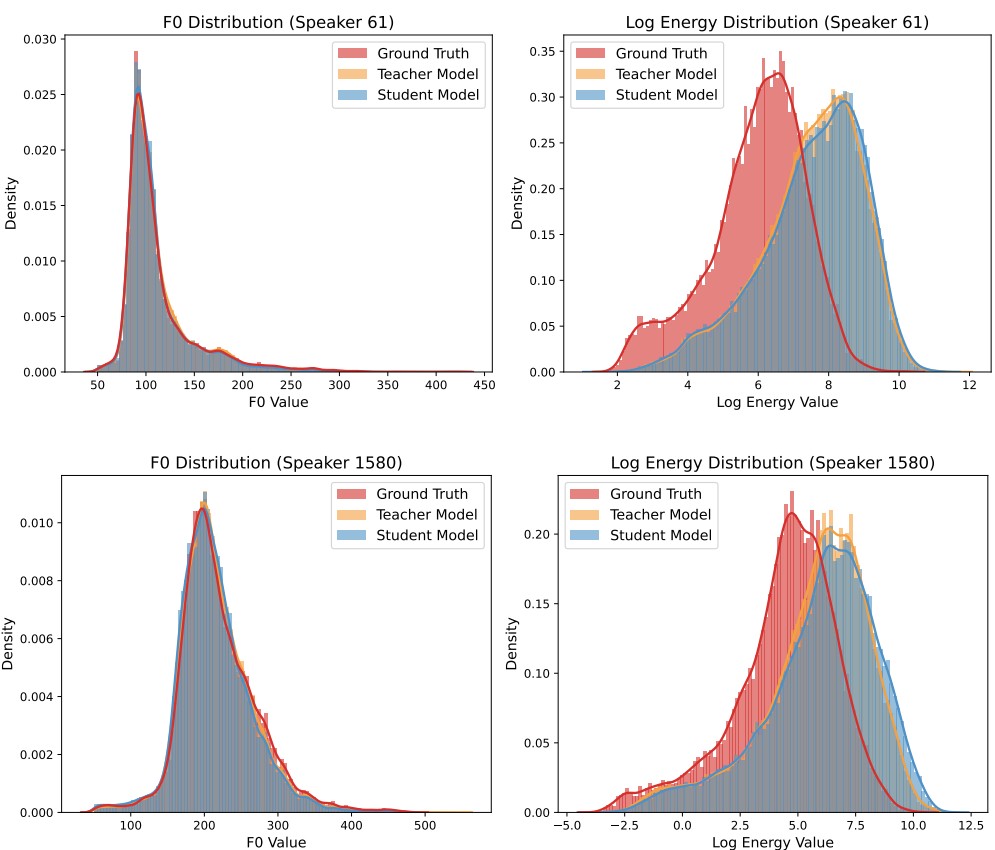

Figure 4: Two examples for mode coverage with continuation task from LibriSpeech *test-clean* subset. The model continues from a prompt with the exact same text as the ground truth. This task synthesizes speech with varying prompts and texts but from the same speaker, allowing us to compare the mode coverage without the same text and prompt. The student exhibits very similar behavior to the teacher and shows minimal mode shrinkage. The misalignment in energy between ground truth and our models is caused by normalization during data pre-processing where the audio is normalized between -1 to 1 in amplitude, causing the generated samples to have a different amplitude range.

We further assessed the model's mode coverage quantitatively by calculating the Wasserstein distance between the student and teacher models, as well as the ground truth, in terms of pitch (F0) and energy. The Wasserstein distances $W_{f_0}$ (for pitch) and $W_N$ (for energy) were computed across all 40 speakers in the LibriSpeech *test-clean* subset. Additionally, we compared the Wasserstein distance between the student and teacher $W(p_\theta, p_\phi)$ in both *conditional* and *unconditional* settings. The conditional case involved synthesizing speech 50 times with the same text and prompt, while the unconditional case used varying texts and prompts from the same speaker as a speech continuation task.

As shown in Table 5, the difference between the student and teacher in terms of Wasserstein distance to the ground truth is relatively small in the unconditional case, and the distance between the student and teacher is much smaller compared to the conditional case (2.55 vs. 16.53). This suggests that the reduction in diversity, or mode shrinkage, primarily occurs in the conditional setting (i.e., when

Table 5: Wasserstein distance between student distribution ($p_\theta$), teacher distribution ($p_\phi$) and real data distribution ($p_{\text{real}}$) when samples are generated with the same text and prompt and varying texts and prompts in terms of pitch (F0) and log energy.

| Sample Conditons | Aspect | $W(p_\theta, p_{\text{data}})$ | $W(p_\phi, p_{\text{data}})$ | $W(p_\theta, p_\phi)$ |
|---|---|---|---|---|
| Varying text-prompt pairs (*unconditional*) | Pitch ($W_{f_0}$) | 3.35 | 2.25 | 2.55 |
| Same text-prompt pairs (*conditional*) | Pitch ($W_{f_0}$) | — | — | 16.53 |
| Varying text-prompt pairs (*unconditional*) | Energy ($W_N$) | 5.47 | 4.88 | 1.34 |
| Same text-prompt pairs (*conditional*) | Energy ($W_N$) | — | — | 12.49 |

synthesizing with the same text and prompt). In the unconditional setting, the student model still spans the entire support of the teacher's distribution and closely matches the ground truth distribution.

Given that zero-shot TTS is highly conditional, where the output must closely match the prompt in both voice and style, this reduction in conditional diversity is not necessarily a drawback. In fact, this narrowing of diversity is often preferred by human listeners, as it leads to outputs that are more aligned with the prompt, as demonstrated in Figure 2 and Table 4.

# B  ADDITIONAL EVALUATION RESULTS

We conducted additional evaluations of acoustic features that capture emotional nuances in speech, following Li et al. (2022), focusing on pitch (mean and standard deviation), energy (mean and standard deviation), Harmonics-to-Noise Ratio (HNR), jitter, and shimmer.

Table 6 compares our model with several baselines. Our model consistently outperforms others across all metrics, except for energy mean, likely due to data normalization during preprocessing, which scales audio between -1 and 1, misaligning the energy with the prompt. Nevertheless, our model's higher scores across other features demonstrate its capability to reproduce the emotional content of the prompt speech effectively.

Table 6: Correlation of acoustic features related to speech emotions between synthesized speech and prompt compared to other baseilne models.

| Model | Pitch mean | Pitch standard deviation | Energy mean | Energy standard deviation | HNR | Jitter | Shimmer |
|---|---|---|---|---|---|---|---|
| DMDSpeech (N=4) | **0.93** | **0.52** | 0.40 | **0.52** | **0.86** | **0.77** | **0.69** |
| Teacher (N=128) | 0.86 | 0.37 | 0.30 | 0.34 | 0.79 | 0.65 | 0.56 |
| DiTTo-TTS | 0.89 | 0.41 | **0.76** | 0.17 | 0.82 | 0.71 | 0.65 |
| VoiceCraft | 0.84 | 0.38 | 0.74 | 0.23 | 0.78 | 0.61 | 0.60 |
| CLaM-TTS | 0.85 | 0.39 | 0.61 | 0.31 | 0.79 | 0.66 | 0.61 |
| XTTS | 0.91 | 0.42 | 0.38 | 0.01 | 0.85 | 0.70 | 0.64 |

In the ablation study presented in Tables in 7, we compare the impact of different training strategies on preserving emotional content in synthesized speech. The teacher model shows strong correlations for most acoustic features, while DMD 2 only models demonstrate performance improvements with additional sampling steps, similar to SIM results in Table 4. Adding CTC loss improves word error rate (WER) but does not significantly enhance speaker-related features. However, including SV loss significantly improves speaker-related features, with the model trained with SV loss only achieving the highest scores in multiple metrics, such as pitch mean (0.94), HNR (0.87), and shimmer (0.65). This highlights the importance of SV loss in capturing speaker identity and emotional content.

Finally, reducing the batch size from 96 to 16 resulted in a slight performance drop across most metrics, demonstrating the importance of maintaining a larger batch size for optimal performance in distribution matching distillation.

Table 7: Correlation of acoustic features related to speech emotions between synthesized speech and prompt for the ablation study. The best-performing model is highlighted while the second best model is underlined.

| Model | Pitch mean | Pitch standard deviation | Energy mean | Energy standard deviation | HNR | Jitter | Shimmer |
|---|---|---|---|---|---|---|---|
| Teacher (N=128) | 0.86 | 0.37 | 0.30 | 0.34 | 0.79 | 0.65 | 0.56 |
| DMD 2 only (N=1) | 0.84 | 0.32 | 0.15 | 0.43 | 0.65 | 0.60 | 0.10 |
| DMD 2 only (N=4) | 0.87 | 0.36 | 0.38 | 0.36 | 0.76 | 0.64 | 0.44 |
| $+\mathcal{L}_{\text{CTC}}$ only | 0.91 | 0.40 | 0.34 | 0.40 | 0.77 | 0.63 | 0.46 |
| $+\mathcal{L}_{\text{SV}}$ only | **0.94** | **0.54** | **0.41** | **0.52** | **0.87** | **0.77** | 0.65 |
| DMDSpeech (N=4) | 0.93 | 0.52 | 0.40 | **0.52** | 0.86 | **0.77** | **0.69** |
| B. S. $96 \rightarrow 16$ | 0.92 | 0.48 | 0.39 | 0.51 | 0.85 | 0.74 | 0.60 |

## C  IMPLEMENTATION DETAILS

### C.1  DAC VARIATIONAL AUTOENCODER

We utilize a latent audio autoencoder to compress raw waveforms into compact latent representations for diffusion modeling. Our architecture follows the DAC model proposed by Kumar et al. (2024), with a key modification to use a variational autoencoder (VAE) bottleneck instead of residual vector quantization, enabling continuous latent spaces and end-to-end differentiable training.

The DAC consists of an encoder $\mathcal{E}$, a VAE bottleneck, and a decoder $\mathcal{D}$. The encoder maps the input waveform $\mathbf{y} \in \mathbb{R}^{1 \times T}$ into a latent representation $\mathbf{x} \in \mathbb{R}^{C \times L}$, where $C$ and $L$ denote channels and downsampled temporal resolution. The VAE bottleneck introduces stochasticity by modeling $\mathbf{x}$ as a distribution, and the decoder reconstructs the waveform by minimizing the reconstruction loss.

The encoder applies an initial convolution followed by residual units with dilated convolutions at scales $1, 3, 9$ to capture multi-scale temporal features. After each block, strided convolutions reduce the temporal resolution by a factor of 1200. For 48 kHz audio, the encoded latent is 40 Hz, making it ideal for efficient speech synthesis tasks. The latent channel dimension of our autoencoder is $C = 64$.

The encoder's output is split into mean $\boldsymbol{\mu}$ and scale $\boldsymbol{\sigma}$ parameters:

$$\mathbf{z} = \boldsymbol{\mu} + \boldsymbol{\sigma} \odot \boldsymbol{\epsilon}, \quad \boldsymbol{\epsilon} \sim \mathcal{N}(\mathbf{0}, \mathbf{I}), \tag{18}$$

where $\mathbf{z}$ is sampled using the reparameterization trick (Kingma, 2013). The decoder mirrors the encoder with transposed convolutions and residual units to upsample latent representations back to the original waveform $\hat{\mathbf{y}} = \mathcal{D}(\mathbf{z})$, where $\hat{\mathbf{y}}$ is the reconstructed waveform. The encoder and decoder architectures are the same as DAC (Kumar et al., 2024).

The KL divergence between the approximate posterior $q(\mathbf{z}|\mathbf{y})$ and prior $p(\mathbf{z})$ is computed as:

$$\mathcal{L}_{\text{KL}} = \mathbb{E}_{\mathbf{y}} \left[ \frac{1}{N} \sum_{i=1}^{N} \left( \mu_i^2 + \sigma_i^2 - \log \sigma_i^2 - 1 \right) \cdot \mathbf{m}_i \right], \tag{19}$$

where $N$ is the number of channels, and $\mathbf{m}_i$ is the channel mask. The autoencoder is trained to minimize a combination of reconstruction loss and KL divergence:

$$\mathcal{L}_{\text{AE}} = \mathbb{E}_{\mathbf{y}} \left[ \|\mathbf{y} - \hat{\mathbf{y}}\|_1 \right] + \lambda_{\text{KL}} \mathcal{L}_{\text{KL}}, \tag{20}$$

where $\lambda_{\text{KL}} = 0.1$ to balance the KL loss. In addition to the KL loss, we also employ adversarial training following Kumar et al. (2024) with the complex STFT discriminator.

### C.2  DMDSPEECH

In this section, we present the implementation details of our DMDSpeech model, including the noise schedule, gradient calculation of DMD loss, detailed architecture, and sampling algorithm.

### C.2.1 SHIFTED COSINE NOISE SCHEDULE

We follow Lovelace et al. (2023); Hoogeboom et al. (2023) and use the shifted cosine noise schedule with $\alpha_t$ and $\sigma_t$ denoting the amount of signal and noise at time $t$. The noise-to-signal ratio (SNR) $\lambda_t = \alpha_t/\sigma_t$ of the noise schedule is shifted by a factor $s$, from which the shifted SNR $\lambda_{t,s}$ and noise schedule $\alpha_{t,s}, \sigma_{t,s}$ are defined:

$$\alpha_t = \cos\left(\frac{\pi}{2}t\right) \qquad (21) \qquad \lambda_{t,s} = \frac{\alpha_{t,s}}{\sigma_{t,s}} = \lambda_t \cdot s^2 = \frac{\alpha_t}{\sigma_t} \cdot s^2, \qquad (22)$$

Using the fact $\alpha_t = \text{sigmoid}\left(\log(\lambda_t)\right)$ as stated in Kingma et al. (2021), the shifted noise schedule can then be computed in the log space for numerical stability:

$$\alpha_{t,s} = \text{sigmoid}\left(\log(\lambda_t) + 2\log(s)\right), \qquad (23) \qquad\qquad \sigma_{t,s} = \sqrt{1 - \alpha_{t,s}^2}. \qquad (24)$$

Lower $s$ emphasizes the higher noise levels and can potentially improve the model's performance. We set $s = 0.5$ following Lovelace et al. (2023) as it is shown to produce the most robust results.

### C.2.2 GRADIENT CALCULATION OF DMD LOSS

The gradient of the DMD loss with respect to the generator parameters $\theta$ is given by eq. 4. The actual implementation of gradient calculation follows the following steps.

We first sample latent variables $\mathbf{x}_t$ are generated via forward diffusion process as:

$$\mathbf{x}_t = \alpha_t \mathbf{x}_0 + \sigma_t \boldsymbol{\epsilon}, \qquad (25)$$

where $\mathbf{x}_0$ is the clean latent representation, and $\boldsymbol{\epsilon} \sim \mathcal{N}(0, \mathbf{I})$.

The clean latents $\hat{\mathbf{x}}_0^{\text{real}}$ and $\hat{\mathbf{x}}_0^{\text{fake}}$ then are estimated using the predicted noise by both of the teacher $f_\phi$ and student $g_\psi$ diffusion models following eq. 7 and eq. 8, respectively.

From there, we calculate the numerical gradient of $\mathcal{L}_{\text{DMD}}$. We define the following quantity as the difference between the ground truth clean latent and estimated latents:

$$p_{\text{real}} = \mathbf{x}_0 - \hat{\mathbf{x}}_0^{\text{real}}, \qquad (26) \qquad\qquad p_{\text{fake}} = \mathbf{x}_0 - \hat{\mathbf{x}}_0^{\text{fake}}. \qquad (27)$$

Then the difference in score $\Delta$ (numerical gradient) can be calculated as:

$$\Delta = \omega_t \alpha_t \left(s_{\text{real}} - s_\theta\right) \qquad (28)$$

$$= \omega_t \alpha_t \left(-\frac{\left(\mathbf{x}_t - \alpha_t \hat{\mathbf{x}}_0^{\text{real}}\right) - \left(\mathbf{x}_t - \alpha_t \hat{\mathbf{x}}_0^{\text{fake}}\right)}{\sigma_t^2}\right) \qquad (29)$$

$$= \omega_t \frac{\alpha_t^2}{\sigma_t^2}\left(-\left(\hat{\mathbf{x}}_0^{\text{real}} - \hat{\mathbf{x}}_0^{\text{fake}}\right)\right). \qquad (30)$$

$$\qquad (31)$$

where the weighting factor $\omega_t$ is defined as:

$$\omega_t = \frac{\sigma_t^2}{\alpha_t \left\|\mathbf{x}_0 - \hat{\mathbf{x}}_0^{\text{real}}\right\|_1} = \frac{\sigma_t^2}{\alpha_t \left\|p_{\text{real}}\right\|_1}. \qquad (32)$$

Hence, eq. 28 can be written as:

$$\Delta = \frac{\left(p_{\text{real}} - p_{\text{fake}}\right)}{\left\|p_{\text{real}}\right\|_1}, \qquad (33)$$

which is back-propagated to $G_\theta$ via gradient descent algorithm.

### C.2.3 DETAILED ARCHITECTURE

In this section, we present the architecture of our Diffusion Transformer (DiT) model (Peebles & Xie, 2023). The DiT model integrates diffusion processes with transformer architectures to generate high-quality speech representations conditioned on textual input.

Our DiT model consists of the following key components:

- Embedding Layers: Transform input IPA tokens, binary prompt masks, and speech latents into continuous embeddings.

- Transformer Encoder: Encodes the textual input (IPA tokens) into contextual representations.

- Transformer Decoder: Decodes the latent representations conditioned on the encoder outputs and additional embeddings.

The model parameters are summarized in Table 8.

Table 8: DMDSpeech DiT model parameters.

| Parameter | Value |
|---|---|
| Latent dimension | 64 |
| Model dimension | 1024 |
| Feed-forward dimension | 3072 |
| Number of attention heads | 8 |
| Number of encoder layers | 8 |
| Number of decoder layers | 16 |
| Feed-forward activation function | SwiGLU |
| Text conditioning dropout | 0.1 |
| Noise schedule shifting scale ($s$) | 0.5 |

The embedding layer maps input tokens and latent variables into continuous embeddings. Specifically, IPA tokens are embedded into vectors of size 1024 using an embedding matrix, and speech latents are projected from dimension 64 to 1024 using a linear layer. A binary mask prompt indicating prompt positions $\mathbf{m}$ in the latent sequence is encoded into a mask embedding, and a sinusoidal time embedding represents the diffusion timestep $t$. Positional embeddings are added to both IPA and latent embeddings to encode positional information.

The encoder processes the embedded IPA tokens through 8 layers, each containing multi-head self-attention and feed-forward sublayers with layer normalization and residual connections. The feed-forward sublayers use a hidden dimension of 3072 and the SwiGLU activation function. The encoder outputs the text condition $\mathbf{c}$.

The decoder generates latent representations conditioned on the encoder outputs and additional embeddings over 16 layers. Each layer includes self-attention, cross-attention with the encoder outputs, and feed-forward sublayers. Adaptive layer normalization (AdaLN), conditioned on the timestep embedding, is applied within the decoder. The output layer projects the decoder outputs back to the latent space dimension of 64 using a linear layer.

Classifier-free guidance (CFG) is employed by randomly dropping the textual conditioning during training with a probability of 0.1 and $\omega$ is the guidance scale. The modified $s_{\text{real}}$ with CFG becomes:

$$s_{\text{real}}(\mathbf{x}_t;\omega) = f_{\phi}(\mathbf{x}_t\,;\mathbf{c},\mathbf{m},t) + \omega\left(f_{\phi}(\mathbf{x}_t\,;\mathbf{c},\mathbf{m},t) - f_{\phi}(\mathbf{x}_t\,;\emptyset,\mathbf{m},t)\right), \quad (34)$$

where $\emptyset$ denotes the null condition of $\mathbf{c}$ which is a fixed embedding. We set $\omega = 2$ both for inference of the teacher model and DMD training.

The teacher model generates samples through DDPM sampler (Ho et al., 2020) with discrete time steps $\{t_i\}_{i=1}^N \subset [0,1]$ where $N$ is the total sampling steps:

$$\mathbf{x}_{n-1} = \frac{1}{\alpha_{t_n}}\left(\mathbf{x}_n - \frac{\sigma_{t_n}^2}{\alpha_{t_n}} f_{\phi}(\mathbf{x}_n\,;\mathbf{c},\mathbf{m},t_n)\right) + \sigma_{t_{n-1}}\boldsymbol{\epsilon}, \quad (35)$$

where $\boldsymbol{\epsilon} \sim \mathcal{N}(0,\mathbf{I})$ if $n > 1$, and $\boldsymbol{\epsilon} = \mathbf{0}$ if $n = 1$.

### C.2.4 DMD SAMPLING

Our sampling algorithm of the student (DMDSpeech) is similar to that of the consistency model (Song et al., 2023). The sampling procedure is outlined in Algorithm 1.

---

**Algorithm 1** DMD Multi-Step Sampling Procedure

---

**Require:**
- $\mathbf{c}$: the text embeddings
- $\mathbf{x}_{\text{prompt}}$: the prompt latent
- $L$: total length of the target speech
- $\{t_i\}_{i=1}^{N}$: noise level schedule with $N$ steps

1: Initialize noisy latent $\mathbf{x}_t \sim \mathcal{N}(0, \mathbf{I})$ of shape $(L, d_{\text{latent}})$
2: **for** $i = 1$ to $N$ **do**
3:      $\mathbf{x}_t \leftarrow \mathbf{x}_t \odot (1 - \mathbf{m}) + \mathbf{x}_{\text{prompt}} \odot \mathbf{m}$                $\triangleright$ Re-apply prompt
4:      $v \leftarrow G_\theta(\mathbf{x}_t; \mathbf{c}, \mathbf{m}, t_i)$                  $\triangleright$ Run student network
5:      $\mathbf{x}_0 \leftarrow \mathbf{x}_t \cdot \alpha_{t_i} - \sigma_{t_i} \cdot v$               $\triangleright$ Predict $\mathbf{x}_0$ from $v$
6:      $\mathbf{x}_0 \leftarrow \mathbf{x}_0 \odot (1 - \mathbf{m}) + \mathbf{x}_{\text{prompt}} \odot \mathbf{m}$      $\triangleright$ Re-apply prompt to $\mathbf{x}_0$
7:      **if** $i < N$ **then**
8:          $\boldsymbol{\epsilon} \sim \mathcal{N}(0, \mathbf{I})$
9:          $\mathbf{x}_t = \alpha_{t_{i+1}} \mathbf{x}_0 + \sigma_{t_{i+1}} \boldsymbol{\epsilon}$      $\triangleright$ Re-noise $\mathbf{x}_0$ to get new $\mathbf{x}_t$ at $t_{i+1}$
10:     **end if**
11: **end for**
12: **return** $\mathbf{x}_0$

---

### C.3 LATENT CTC-BASED ASR MODEL

To directly optimize word error rate (WER) within our speech synthesis framework, we implement a Connectionist Temporal Classification (CTC)-based ASR model that operates on latent speech representations. Traditional ASR models work on raw audio or mel-spectrograms, adding computational overhead and potential mismatches when integrated with latent-based synthesis since we need to decode the latent back into waveforms before computing the ASR output. Our latent ASR model processes these representations directly, enabling efficient, end-to-end computation of the CTC loss and direct WER optimization.

The ASR model is based on the Conformer architecture (Gulati et al., 2020), which effectively captures local and global dependencies using convolution and self-attention. Input latent representations $\mathbf{z} \in \mathbb{R}^{T \times d}$ are processed through a 6-layer conformer stack and the model outputs a logit for each latent token over IPA phonemes.

The ASR model is trained using the CTC loss, allowing alignment-free training of sequence-to-sequence models. The CTC loss is defined using softmax function:

$$\mathcal{L}_{\text{CTC}} = -\log p\left(\mathbf{y} \mid \mathbf{o}\right), \tag{36}$$

where $\mathbf{y}$ is the target IPA sequence, $\mathbf{o}$ represents the logits over the IPA symbols, and $p\left(\mathbf{y} \mid \mathbf{o}\right)$ is computed by summing over all valid alignments between the input and target sequences. The probabilities are calculated as:

$$p_{\pi_t}(t) = \frac{\exp\left(o_{t, \pi_t}\right)}{\sum_{k=1}^{V} \exp\left(o_{t, k}\right)}. \tag{37}$$

We trained our ASR model on CommonVoice (Ardila et al., 2019) and LibriLight (Kahn et al., 2020) datasets for 200k steps with the AdamW (Loshchilov & Hutter, 2018) optimizer. The optimizer configuration is the same as teacher training described in Section 4.1.

### C.4 LATENT SPEAKER VERIFICATION MODEL

We develop a latent speaker verification (SV) model that operates directly on latent speech representations in order to optimize speaker similarity within our speech synthesis framework. Unlike traditional SV models, which process raw audio waveforms, our latent SV model integrates seamlessly with our latent-based synthesis, enabling efficient, end-to-end computation of speaker verification loss for direct speaker similarity optimization.

Our latent SV model fine-tunes our CTC-based ASR model for feature extraction following (Cai & Li, 2024) and integrates it with an ECAPA-TDNN architecture (Desplanques et al., 2020) for

speaker embedding extraction. We train the latent SV model using a distillation approach, transferring knowledge from two pre-trained teacher models: a ResNet-based SV model [5] from the WeSpeaker (Wang et al., 2023b) and EPACA-TDNN with a fine-tuned WavLM Large model [6] as the feature extractor. The training objective minimizes the cosine similarity loss between embeddings from the latent SV model and the concatenated embeddings from the teacher models:

$$\mathcal{L}_{\text{SV}} = \mathbb{E}_{\mathbf{z},\mathbf{y}} \left[ 1 - \frac{\mathbf{e}_{\text{teacher}} \cdot \mathbf{e}_{\text{latent}}}{\|\mathbf{e}_{\text{teacher}}\| \, \|\mathbf{e}_{\text{latent}}\|} \right], \tag{38}$$

where $\mathbf{e}_{\text{latent}}$ and $\mathbf{e}_{\text{teacher}}$ are the embeddings from the latent SV and teacher models, respectively.

Our latent SV model was trained on CommonVoice (Ardila et al., 2019) and LibriLight (Kahn et al., 2020) datasets for 400k steps with the AdamW optimizer. Since we did not use VoxCeleb dataset that was used originally to train the teacher SV models, we used data augmentation [7] to shift the pitch of the speakers to create new speaker identity to prevent overfitting during training.

## D HUMAN RATING CORRELATIONS

We generated scatter plots to visualize the relationships between the four subjective metrics: MOS-N (naturalness), MOS-Q (sound quality), SMOS-V (voice similarity), and SMOS-S (style similarity), and two objective evaluation metrics: word error rate (WER) and speaker embedding cosine similarity (SIM). The scatter plots are displayed in Figure 5, and they cover all subjective evaluation experiments conducted in this work at the utterance level.

Despite the noise and variance in the utterance-level subjective ratings, the plots reveal important trends. A strong correlation exists between human-rated speaker similarity (SMOS-V and SMOS-S) and the SIM score from the speaker verification model, with correlation coefficients of 0.55 and 0.50, respectively. This highlights the alignment between subjective human judgments and the objective speaker embedding similarity. On the other hand, there is a weaker but still significant negative correlation between WER and both naturalness (MOS-N) and sound quality (MOS-Q), with coefficients of $-0.16$ for both. These findings validate our approach to directly optimize these metrics. Future research could explore other differentiable metrics or reward models that align even more closely with human auditory preferences.

## E EVALUATION DETAILS

### E.1 BASELINE MODELS

This section briefly introduces the baseline models used in our evaluations and the methods employed to obtain the necessary samples.

- **CLaM-TTS**: CLaM-TTS (Kim et al., 2024) is a strong autoregressive baseline for zero-shot speech synthesis, trained on various datasets including Multilingual LibriSpeech (MLS) (Pratap et al., 2020), GigaSpeech (Chen et al., 2021), LibriTTS-R (Koizumi et al., 2023), VCTK (Yamagishi et al., 2019), and LJSpeech (Ito & Johnson, 2017). Since this model is not publicly available, we obtained 3,711 samples from the authors using instructions provided by the authors at `https://github.com/keonlee9420/evaluate-zero-shot-tts`.

- **DiTTo-TTS**: DiTTo-TTS (Lee et al., 2024) is a previous state-of-the-art (SOTA) end-to-end model for zero-shot speech synthesis, trained on the same datasets as CLaM-TTS, with the addition of Expresso (Nguyen et al., 2023). Like CLaM-TTS, this model is also not publicly available, so we acquired the same set of 3,711 samples from the authors.

- **NaturalSpeech 3**: NaturalSpeech 3 (Ju et al., 2024) is a previous SOTA model in zero-shot speech synthesis, trained on LibriLight (Kahn et al., 2020). Using factorized codec and

---

[5] Available at `https://huggingface.co/pyannote/wespeaker-voxceleb-resnet34-LM`
[6] `https://github.com/microsoft/UniSpeech/tree/main/downstreams/speaker_verification`
[7] `https://github.com/facebookresearch/WavAugment`

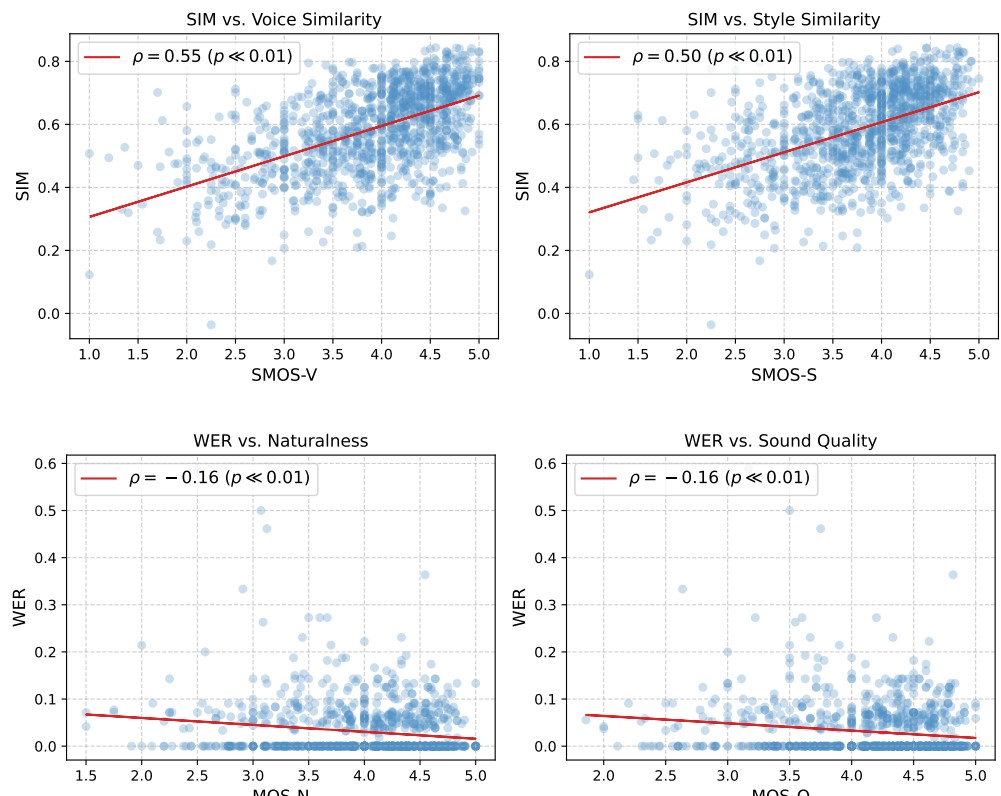

Figure 5: Top: Scatter plots showing the relationship between human-rated naturalness (MOS-N) and sound quality (MOS-Q) versus word error rate (WER). The correlation coefficients are -0.16 for both, indicating a weak negative correlation ($p \ll 0.01$). Bottom: Scatter plots of human-rated voice similarity (SMOS-V) and style similarity (SMOS-S) versus speaker embedding cosine similarity (SIM). The correlation coefficients are 0.55 and 0.50, reflecting a strong positive correlation ($p \ll 0.01$). These plots demonstrate how objective evaluations (WER and SIM) align with subjective human ratings.

discrete diffusion models, it achieves near-human performance in prompt speaker similarity. Since it is not publicly available, we collected 40 samples from the authors, along with text transcriptions and 3-second prompt speeches, to synthesize speech for comparison. We also sourced 7 official samples from `https://www.microsoft.com/en-us/research/project/e2-tts/` tested on the LibriSpeech *test-clean* subset, totally 47 samples.

- **StyleTTS-ZS**: StyleTTS-ZS (Li et al., 2024b) is another previous SOTA model for zero-shot speech synthesis, known for its fast inference speed and high naturalness and speaker similarity. As the model is not publicly available, we requested 47 samples from the authors to match those provided by Ju et al. (2024).

- **VoiceCraft**: VoiceCraft (Peng et al., 2024) is a strong autoregressive baseline model trained on GigaSpeech (Chen et al., 2021) and LibriLight (Kahn et al., 2020), performing well in speaker similarity and can be used for speech editing. This model is publicly available at `https://github.com/jasonppy/VoiceCraft`, and we synthesized 3,711 samples using the same text and 3-second speech prompts provided for CLaM-TTS and DiTTo-TTS with the 830M TTS-enhanced model.

- **XTTS**: XTTS (Casanova et al., 2024) is another strong zero-shot speech synthesis baseline, trained on various public and proprietary datasets totaling around 17k hours. The model is publicly available at `https://huggingface.co/coqui/XTTS-v2`, and we synthesized the same 3,711 samples as above.

## E.2 SUBJECTIVE EVALUATION

Figure 6: Screenshot of the subjective evaluation survey used for the perceptual quality assessment of speech synthesis models. Participants are presented with a reference (prompt) and sample to be evaluated and are asked to rate various attributes such as naturalness, voice similarity, style similarity, and quality on a scale from 1 to 5. If the sample is unintelligible, participants must mark it as "Yes" under the "Is content broken?" section. The survey prevents submission if any slider remains at the default "N/A" position, ensuring that each aspect is rated.

We conducted two subjective evaluations using the Prolific crowdsourcing platform [8] to assess the perceptual quality of the generated speech samples. These evaluations measured key attributes including naturalness, voice similarity, style similarity, and audio quality based on a reference speech sample provided to the raters.

Because some workers may "game" the systems by answering randomly, or skipping the reference sample, we used two forms of validation tests. The first uses mismatched speaker where the test presents the workers with different voices for the reference and test sample, both being real speakers. If a participant rated these mismatched samples with a speaker similarity score above 3, all their ratings were excluded from the analysis. The second validation test involved identical sample pairs, where participants were asked to rate identical reference and sample pairs. If any of the subjective attributes, including naturalness, similarity, style, or quality, were rated below 4 for these identical pairs, all responses from that participant were excluded.

The first subjective evaluation experiment, referred to as the "bigger" experiment, involved 501 unique workers. There are a total of 80 parallel utterances for each method, which include all end-to-end (E2E) baselines and models in the ablation study were rated. The results were present in Table 2 and 4. Each worker was assigned provide ratings for 30 samples. There are 4 validation tests in this experiment. Approximately 30% of the responses were invalidated due to participants failing the validation test at least once. The second, "smaller" experiment that compared non-E2E baselines over 47 utterances per method. There are 290 unique workers, with each worker completing 28 ratings. The validation test is doubled to 8 per test. In this smaller study, 40% of the ratings were invalidated because the stricter validation process led to more failures. The number of invalid samples are consistent with prior work carried out on similar platforms.

---

[8] https://www.prolific.com/

The survey (Figure 6) interface presented participants with a reference (prompt) and a corresponding sample recording. Participants rated each sample on a scale from 1 to 5 across several categories:

1. Naturalness, evaluating how real or synthetic the voice sounded;
2. Quality, determining whether the audio quality was maintained or degraded compared to the prompt;
3. Voice similarity, assessing how closely the sample matched the reference speaker;
4. Style similarity, considering the alignment of the speaking style and emotion;
5. Intelligibility, for which raters were asked to mark it as such to flag broken samples during the analysis if the audio sample was entirely unintelligible.

The last rating category "is the content broken¿' helps us to identify if any samples are unintelligible which would indicate completely failed generation or corrupted files. In the end, we do not have any samples that are rated "broken" by the majority.

Compensation for both experiments is set to a rate of $15 per hour, higher than Prolific's recommendation of $12 per hour with a target average time of 12 minutes per test.

