# OpenReview forum: "DMDSpeech: Distilled Diffusion Model Surpassing The Teacher in Zero-shot Speech Synthesis via Direct Metric Optimization"
_ICLR.cc/2025/Conference — ICLR 2025 Conference Withdrawn Submission_

### Official Review · Reviewer_QWFW · 2024-10-29

**Soundness:** 3
**Presentation:** 3
**Contribution:** 2
**Rating:** 5
**Confidence:** 4

**Summary:**

This paper proposes a distilled TTS diffusion model using CTC and SV loss. While the paper is mostly well-written, it lacks novelty and includes some unfair comparisons. Firstly, distilled diffusion models are already well established in computer vision, with a substantial body of research. Even in audio and speech processing, the authors are not among the first to apply this approach, as seen in Nvidia’s blog [1] and other studies. Using SV loss for TTS is also not novel, as it has been previously applied in [3]. Thus, the primary novelty appears to lie in combining these techniques.
The claim of “DIRECT METRIC OPTIMIZATION” is misleading. In speech synthesis, there is no truly direct metric for evaluating generated speech quality. At best, we rely on MOS, while WER and speaker similarity measures serve only as indirect proxies. Generated speech may exhibit machine-like articulation yet achieve low WER. Additionally, speaker similarity measures can be highly domain-dependent and often perform poorly on out-of-domain speakers.
The comparisons made are not entirely fair, given that training with CTC and SV loss naturally results in improved WER and speaker similarity. For RTF evaluation, it seems the authors mainly compare their model with autoregressive models such as XTTS and VoiceCraft, which are not known for their speed. Moreover, the RTF values for DiTTO-TTS and CLaM-TTS are drawn from their respective papers, which is inaccurate since RTF must be measured on the same hardware to be comparable.
NaturalSpeech 3 is not open-source, making it difficult to assess whether its RTF in this paper reflects the original results. A more appropriate comparison would be with StyleTTS2, an open-source, state-of-the-art non-autoregressive model.


[1]https://developer.nvidia.com/blog/speeding-up-text-to-speech-diffusion-models-by-distillation/
[2]Bai, Yatong, et al. "Consistencytta: Accelerating diffusion-based text-to-audio generation with consistency distillation." arXiv preprint arXiv:2309.10740 (2024).
[3]E. Casanova, J. Weber, C. D. Shulby, A. C. Junior, E. Golge, and ¨ M. A. Ponti, “Yourtts: Towards zero-shot multi-speaker tts and zero-shot voice conversion for everyone,” in International Conference on Machine Learning. P

**Strengths:**

The paper is well written. The experiments are comprehensive although some are not fair comparison.

**Weaknesses:**

The novelty is limited and the claim of directly optimization is simply not true. Some comparisons in experiments section is not fair.

**Questions:**

Have the authors try other objective than CTC?

---

> ### Author Response · Authors · 2024-11-16
> **Point-to-Point Rebuttal to Reviewer QWFW**
>
> We thank the reviewer for acknowledging the strengths of our work, including its clear presentation and comprehensive experiments. These aspects underscore the rigor and reproducibility of our approach. Below, we address the concerns raised regarding novelty and comparisons.
>
> **Response to Novelty Concerns**
>
> While we agree that individual components like distillation and metric optimization have been explored before, we respectfully emphasize several key innovations in our work:
>
> **Regarding Diffusion Model Distillation in TTS:** Previous distillation approaches in audio synthesis consistently showed performance degradation compared to their teacher models. For instance, the work cited by the reviewer [2] reported a MOS decline from 4.136 (teacher) to 3.902 (student). In these studies, the gains only focus on improving the inference speed, not performance. In contrast, our work demonstrates that, through direct metric optimization applied on a distilled diffusion model, it is possible to not only accelerate inference but also to surpass the teacher model in both quality (MOS) and speaker similarity, which has not been demonstrated by previous TTS distillation approaches.
>
> **In Terms of Metric Optimization:**
>
>   1. **True End-to-End Metric Optimization:** While prior works like YourTTS used speaker embedding similarity loss, their approach was fundamentally limited by non-differentiable components (e.g., monotonic alignment search) that prevented gradient flow to crucial components like the text encoder and duration predictor. Our framework enables true end-to-end optimization across all components, resulting in substantial improvements in both objective metrics (SIM, WER) and human evaluations (SMOS-V, MOS). On the other hand, YourTTS did not present consistent and significant improvements over speaker similarity with their proposed SCL loss, suggesting that non-end-to-end optimization is less effective.
>
>   2. **General Framework for Metric Optimization:** Moreover, our framework is more general than just SV loss, as it enables optimization of any differentiable metric. The successful integration of CTC loss demonstrates this generality, something not previously achieved in speech synthesis models due to architectural limitations. This is likely because previous methods without iterative sampling (such as YourTTS) cannot optimize the text encoder and duration predictor simultaneously with the decoder for the CTC loss, components critical for natural and intelligible speech. On the other hand, our use of a distilled diffusion model enables this end-to-end optimization approach, which has proven effective in enhancing both speaker similarity (SIM) and intelligibility (WER) measures, as well as subjective metrics such as SMOS and MOS by human raters.
>
> **Response to Concerns Over “Direct Metric Optimization” Claims**
>
> We understand the reviewer’s perspective on the indirect nature of metrics like SIM and WER. While these metrics do serve as proxies, they are nonetheless strongly correlated with human judgment, as supported by our subjective evaluations (from 290 raters) and correlations demonstrated in Figure 5. Furthermore, we observed that optimizing SIM and CTC loss improves corresponding human-rated metrics, such as SMOS and MOS, as shown in Table 4. We also encourage the reviewer to listen to the effect of direct metric optimization on our demo page: https://dmdspeech.github.io/demo/#ablation. Since we explicitly state that we are optimizing evaluation metrics rather than directly optimizing for subjective human preference, we believe our description is appropriately cautious and accurate.
>
> Moreover, the essence of our direct metric optimization lies in making previously non-optimizable metrics, such as speaker similarity and WER, directly optimizable within an end-to-end framework. This advantage over existing TTS models is meaningful, resulting in practical improvements in quality and alignment with human perception. Hence, we believe that our experiment designs are fair, as our method revolves around the advantages and gains of directly optimizing evaluation metrics.

---

> > ### Author Response · Authors · 2024-11-16
> > **Point-to-Point Rebuttal to Reviewer QWFW (Continued)**
> >
> > **Concerns over RTF Comparision**
> >
> > We recognize the reviewer’s concern about hardware variability in RTF comparisons. Since most state-of-the-art TTS models are not publicly available, we followed standard practice [1,2,3] by reporting RTF values as stated in the original publications for models not publicly available.
> >
> > We also thank the reviewer for suggesting a comparison to StyleTTS 2. While it is not a large-scale zero-shot model comparable to the scope of this paper, we can offer an RTF comparison if needed. For reference, StyleTTS 2 has an RTF of 0.0684, while our model achieves an RTF of 0.0656 on a single Tesla V100. Our RTF calculation used the total duration of all samples in the LibriSpeech test-clean set, providing an average-based RTF.
> >
> > However, it is worth noting that some papers calculate RTF based on synthesizing fixed durations (e.g., 10 seconds) and dividing by that duration [1, 3]. Following this approach, our RTF would be 0.0276, due to parallelism in GPU computations (synthesizing 3 seconds of audio takes approximately the same amount of time as 10 seconds of audio, but the denominator is smaller). We would appreciate further suggestions from the reviewer on a fairer evaluation of RTF compared to other models.
> >
> >
> > **References:**
> >
> > [1] Lee, K., Kim, D. W., Kim, J., & Cho, J. (2024). DiTTo-TTS: Efficient and Scalable Zero-Shot Text-to-Speech with Diffusion Transformer. arXiv preprint arXiv:2406.11427.
> >
> > [2] Ye, Z., Ju, Z., Liu, H., Tan, X., Chen, J., Lu, Y., ... & Guo, Y. (2024, October). Flashspeech: Efficient zero-shot speech synthesis. In Proceedings of the 32nd ACM International Conference on Multimedia (pp. 6998-7007).
> >
> > [3] Chen, Y., Niu, Z., Ma, Z., Deng, K., Wang, C., Zhao, J., ... & Chen, X. (2024). F5-TTS: A Fairytaler that Fakes Fluent and Faithful Speech with Flow Matching. arXiv preprint arXiv:2410.06885.

---

### Official Review · Reviewer_9zdq · 2024-10-29

**Soundness:** 2
**Presentation:** 3
**Contribution:** 2
**Rating:** 3
**Confidence:** 4

**Summary:**

This paper introduces DMDSpeech, a distilled diffusion model designed for efficient, high-quality zero-shot speech synthesis. It optimizes perceptual metrics by incorporating Connectionist Temporal Classification (CTC) and Speaker Verification (SV) losses, targeting improvements in word error rate (WER) and speaker similarity. The model outperforms previous state-of-the-art approaches in naturalness and speaker similarity, while achieving faster synthesis. This approach highlights the benefits of direct metric optimization in TTS and demonstrates the effectiveness of DMDSpeech in aligning generated speech with human auditory preferences.

**Strengths:**

**Integration of CTC and SV losses** The paper optimize diffusion TTS model by using end-to-end metric optimization, applying CTC and SV losses to improve word error rate (WER) and speaker similarity.

**Model results** DMDSpeech achieves better performance compared to state-of-the-art baselines with significantly reduced inference time.

**Weaknesses:**

**Limited Novelty of the Proposed Approach** DMDSpeech utilizes the existing DMD 2 method to distill the teacher model for faster sampling and direct metric optimization, except that DMD 2 simulates a four-step inference process during training, while DMDSpeech simulates a single step. Overall, this paper feels more like an application of DMD 2 in text-to-speech rather than an original methodological advancement.

**Need for Additional Experimental Validation** In DMDSpeech, the student generator is trained to match the teacher model's distribution via distribution matching distillation, but the results in Table 4 indicate that DMD 2 alone decreases MOS-N, SMOS, SMOS-S, and SIM scores compared to the teacher model, with MOS-Q remaining similar. This suggests DMD 2 may offer limited improvement. Why not simply fine-tune the pretrained teacher model with multi-modal adversarial learning and direct metric optimization? Could you conduct additional experiments comparing the teacher model with L_CTC and L_SV​ using GAN against DMDSpeech? Also, please compare the teacher model with L_CTC and L_SV without GAN against DMDSpeech. These would help clarify the impacts of DMD, DMD 2, and GAN in DMDSpeech.

**Questions:**

1. In line 412, could you elaborate on the statement, "Table 5 shows that our model achieved the highest speaker similarity score (SIM) to the prompt, even surpassing the ground truth"? It is unclear how Table 5 demonstrates that DMDSpeech exceeds the ground truth.

2. Does your training dataset exclude the official samples from the baseline models which are used for evaluation?

3. Could you clarify why you chose to fine-tune the CTC-based ASR model to derive the latent SV model? From your explanation in Appendix C.4, you use a distillation approach to align the CTC-based ASR model's embeddings with the concatenated embeddings from the WeSpeaker's ResNet-based SV model and EPACA-TDNN with a fine-tuned WavLM Large model. Wouldn't it be more efficient to utilize the WeSpeaker's ResNet-based SV model and EPACA-TDNN with a fine-tuned WavLM Large model to directly extract the concatenated embeddings of the prompt and the concatenated embeddings of the generated speech, then calculate the SV loss in Equation 14 using these two embeddings? Can you provide experiment to evaluate that the latent SV model fine-tuned from CTC-based ASR model performs better in speaker similarity?

**Details Of Ethics Concerns:**

Research involving human subjects

---

> ### Author Response · Authors · 2024-11-16
> **Point-to-Point Rebuttal to Reviewer 9zdq**
>
> We appreciate the reviewer’s feedback for recognizing the strong results and advantages of DMDSpeech, including its superior performance in naturalness, speaker similarity, and inference speed. We appreciate the reviewer’s feedback and questions. Below, we address each point with clarifications and additional details.
>
> **Response to Concerns about Limited Novelty**
>
> We agree that DMD 2 is an important component of our work, but we respectfully clarify that it is not our only contribution. While we do utilize DMD 2, it serves as a foundational component to achieve our primary objective: creating an efficient TTS model with high-quality output and fast inference.
>
> Our primary innovation lies in combining DMD 2 with direct metric optimization to achieve both faster inference and better performance than existing approaches. As demonstrated in Table 4, applying DMD 2 alone does not improve performance beyond the teacher model except for inference speed. Significant improvements in naturalness and speaker similarity have been made in our novel application of direct metric optimization. We encourage the reviewer to listen to the audio samples on our demo page (https://dmdspeech.github.io/demo/#ablation) to hear the clear quality improvements achieved through direct metric optimization.
>
> Our methodological contribution is significant because:
> 1. We enable end-to-end direct metric optimization for TTS, which was not previously possible due to non-differentiable components (e.g., duration predictor or monotonic alignment search) or iterative sampling (e.g., autoregressive models or diffusion models) in existing models. By distilling the diffusion model without non-differentiable components into a single-step generator, we overcome these obstacles, making it possible to directly optimize differentiable metrics.
>
> 2. We demonstrate that this approach can surpass teacher model performance in both speed and quality.
>
> 3. Our framework is generalizable: as evaluation metrics continue to evolve, DMDSpeech’s approach allows for future improvements in human preference alignment. Therefore, we believe our work offers a novel methodology and significant advancement in TTS, as acknowledged by Reviewer i2SD.
>
> **Response to Request for Additional Experiments**
>
> We appreciate the reviewer’s suggested experiments but wish to clarify a fundamental aspect of diffusion models that limits the possibility of these suggested experiments: Diffusion-based models do not produce the final output directly; instead, they compute intermediate “score” (added noise) estimates at each step of the denoising process. While theoretically possible, retrieving the final output from a high-noise level yields unintelligible audio, making direct metric optimization infeasible.
>
> To illustrate this limitation, we have added samples at various noise levels to our demo page: https://dmdspeech.github.io/demo/#rebuttal. These samples demonstrate why applying direct metric optimization to the teacher model without iterative sampling (which would require tens to hundreds of steps) is impractical. The speech is unintelligible and speaker identity is unclear at higher noise levels, making it impossible to derive useful gradients from SV or CTC losses. While it is possible to sample through multiple steps to generate the final output, backpropagating across multiple steps in a large model also leads to instability and high computational costs, making the suggested experiments impractical.
>
> Thus, while we value the input, we would like the reviewer to understand that the experiments proposed do not align with the operational structure of diffusion-based models in TTS.

---

> ### Author Response · Authors · 2024-11-16
> **Point-to-Point Rebuttal to Reviewer 9zdq (Continued)**
>
> **Additional Clarifications**
>
> >In line 412, could you elaborate on the statement, "Table 5 shows that our model achieved the highest speaker similarity score (SIM) to the prompt, even surpassing the ground truth"? It is unclear how Table 5 demonstrates that DMDSpeech exceeds the ground truth.
>
> Thank you for catching the table reference error. This was indeed a typo. We meant to reference Table 3, not Table 5, regarding our model achieving the highest SIM score. We will correct this in the revision.
>
> >Does your training dataset exclude the official samples from the baseline models which are used for evaluation?
>
> Yes, our training dataset (LibriLight) contains no speakers and utterances from the LibriSpeech test dataset. As stated by Kahn et al. (2020) in the LibriLight dataset paper: "We then removed corrupted files, files with unknown or multiple speakers, and speakers appearing in LibriSpeech dev and test sets."
>
> >Could you clarify why you chose to fine-tune the CTC-based ASR model to derive the latent SV model? From your explanation in Appendix C.4, you use a distillation approach to align the CTC-based ASR model's embeddings with the concatenated embeddings from the WeSpeaker's ResNet-based SV model and EPACA-TDNN with a fine-tuned WavLM Large model. Wouldn't it be more efficient to utilize the WeSpeaker's ResNet-based SV model and EPACA-TDNN with a fine-tuned WavLM Large model to directly extract the concatenated embeddings of the prompt and the concatenated embeddings of the generated speech, then calculate the SV loss in Equation 14 using these two embeddings? Can you provide experiment to evaluate that the latent SV model fine-tuned from CTC-based ASR model performs better in speaker similarity?
>
> We appreciate this insightful question. Our work follows recent developments in diffusion-based TTS models that use the latent diffusion framework, which operates on latent representations rather than waveforms. Since both the WeSpeaker ResNet-based SV model and EPACA-TDNN with WavLM Large were trained on waveform inputs, they cannot be directly applied to our latent representations. Converting these latents back to waveforms during training would be both inefficient and computationally intractable due to GPU memory constraints. Therefore, we distilled these waveform-based SV models into latent SV models for both efficiency and applicability. We will clarify this technical constraint in our revision.
>
> **Reference:**
>
> Kahn, J., Riviere, M., Zheng, W., Kharitonov, E., X*u, Q., Mazaré, P. E., ... & Dupoux, E. (2020, May). Libri-light: A benchmark for asr with limited or no supervision. In ICASSP 2020-2020 IEEE International Conference on Acoustics, Speech and Signal Processing (ICASSP) (pp. 7669-7673). IEEE.

---

### Official Review · Reviewer_4Zge · 2024-11-01

**Soundness:** 4
**Presentation:** 4
**Contribution:** 2
**Rating:** 3
**Confidence:** 3

**Summary:**

In this work, the authors introduce DMDSpeech, a distilled diffusion-based TTS model. In this framework, a student model distills the distribution learned by a diffusion-based teacher model, enabling the generation of high-quality speech in just four steps. By integrating the distillation loss with speaker verification (SV) and connectionist temporal classification (CTC) losses, DMDSpeech achieves excellent speaker similarity and a low word error rate while maintaining high speech quality. The experimental results demonstrate that DMDSpeech outperforms several state-of-the-art baselines as well as the teacher diffusion model.

**Strengths:**

- This paper is technically sound and well-presented.
- It provides ample implementation details, allowing readers to reproduce the results effectively.
- The authors conduct sufficient experiments to showcase the advantages of diffusion matching distillation, such as faster generation times, and highlight the effectiveness of combining speaker verification (SV) and connectionist temporal classification (CTC) perceptual losses.
- Regarding the mode shrinkage phenomenon resulting from the distillation process, the authors offer a detailed analysis, explaining that this effect may not be undesirable in the context of unconditional TTS.

**Weaknesses:**

The two major themes of this paper—(1) distillation of diffusion-based TTS models and (2) joint optimization of TTS and perceptual losses—have both been extensively explored in the field of speech generation. While the paper is well-presented and technically sound, the proposed method is relatively straightforward and lacks novelty.

**Questions:**

The paper is clear enough.

---

> ### Author Response · Authors · 2024-11-16
> **Point-to-Point Rebuttal to Reviewer 4Zge**
>
> We sincerely appreciate your review and valuable feedback on our paper. We are pleased that you found our work technically sound and well-presented, with sufficient implementation details and experiments showcasing the advantages of our approach. We would like to clarify the novelty and distinct contributions of our approach:
>
> 1. **Direct Metric Optimization in TTS:**
> Our work is the first to enable direct, end-to-end optimization of key metrics such as speaker similarity (SIM) and word error rate (WER) in zero-shot TTS. Traditional models like YourTTS and NaturalSpeech 2 & 3 cannot achieve this due to non-differentiable components (e.g., monotonic alignment search, duration predictors) that block gradient flow, making optimization of text encoder and duration predictor for metrics such as WER with CTC loss ineffective.
>
> 2. **Challenges in Iterative Sampling Models:**
> Recent approaches (e.g., Vall-E, DiTTo-TTS) rely on iterative sampling, requiring up to 100+ steps in autoregressive models or 16 steps in flow-matching diffusion models. Backpropagation through these steps is computationally prohibitive and unstable.
>
> 3. **Our Solution:**
> By distilling the diffusion model into a single-step generator, we overcome these limitations, enabling efficient and stable optimization of differentiable metrics. This results in consistent improvements over the teacher model across MOS, SMOS, WER, and SIM. Our originality and novel contribution are also acknowledged by Reviewer i2SD.
>
> We hope this clarifies the meaningful advancements of our work. We also invite any further questions or specific examples of comparable approaches. Thank you for your valuable feedback.

---

> > ### Comment · Reviewer_4Zge · 2024-11-20
> >
> > As mentioned in the paper, many previous TTS frameworks (not necessarily diffusion models) have explored the direct optimization of metrics such as MOS. Therefore, I find the direct optimization of SV and CTC-WER metrics proposed in this work to be relatively straightforward, with the core idea remaining similar to previous TTS frameworks that focus on direct metric optimization.
> >
> > At the same time, a lot of efforts have been made in previous research to address the inefficiency of the iterative sampling process in diffusion-based TTS models, such as through consistency distillation and progressive distillation. This work introduces another solution, distribution matching distillation (DMD), which has previously been tested on image diffusion models. However, the motivation and core concept of DMD do not appear to differ significantly from those of other distillation-based TTS models.
> >
> > I agree that direct metric optimization can be challenging for generative models that rely on multi-step iterative sampling. Therefore, the primary contribution of this work lies in combining two popular enhancements to diffusion-based TTS models: distillation and direct metric optimization. However, aside from this combination, no fundamentally new ideas or concepts are introduced. As a result, I consider the lack of novelty to be the main weakness of this paper.

---

> ### Author Response · Authors · 2024-11-21
>
> We sincerely appreciate the reviewer's continued engagement and feedback on our work. Below, we address your follow-up concerns and provide further clarification.
>
> **Response to "Direct Metric Optimization" as a Straightforward Idea**
>
> As far as we know, no prior TTS paper has implemented **true end-to-end direct metric optimization** with significant improvements across key metrics such as SIM, WER, and MOS. For instance:
>
>    - YourTTS included a speaker similarity loss (SCL) but explicitly reported minimal impact, stating: `According to the Sim-MOS, the use of SCL did not bring any improvements; however, the confidence intervals of all experiments overlap, making this analysis inconclusive.`
>    - Metrics like WER require optimization of the text encoder and duration predictor simultaneously, which is infeasible in prior TTS frameworks with non-differentiable components such as monotonic alignment search (MAS).
>
> Our method overcomes these challenges by enabling true end-to-end optimization for metrics like WER through CTC loss, a novel contribution in the speech synthesis domain. To our knowledge, this is the first approach that achieves such significant and consistent improvements, which we kindly ask the reviewer to acknowledge or provide references to similar works, if they exist.
>
> **Response to "DMD Motivation Similar to Other Distillation Approaches"**
>
> We respectfully disagree that the application of DMD to speech synthesis is trivial. Speech synthesis presents unique challenges compared to the image domain, as it requires strong conditional generation rather than the weak conditional generation typical in image synthesis. This fundamental difference is analyzed in detail in Section 3.4. We have also provided new insights into mode shrinkage in strong conditional generation and its potential benefit and drawback in Section 4.4 and Appendix A, a phenomenon not previously explored in speech synthesis with diffusion models. These contributions have also been acknowledged by Reviewer i2SD.
>
> Furthermore, we emphasize that while prior distillation approaches like progressive or consistency distillation focused solely on improving inference speed, our method achieves **both speed improvements and performance gains over the teacher model**, something not previously reported in the TTS field. This makes our motivation completely different from previous works.
>
> **Response to "No Fundamentally New Ideas"**
>
> We respectfully contend that our contributions are both novel and impactful:
>
> 1. **End-to-End Direct Metric Optimization:** No previous method has successfully achieved this for metrics like WER or SIM, which depend on optimizing all components used for speech synthesis, and our optimization of WER using CTC loss is the first of its kind. Our approach outperforms state-of-the-art models and even the teacher model, demonstrating the practical utility of this innovation.
>
> 2. **General Framework:** Our method is applicable to **any different metric**, including MOSNet, and can adapt as evaluation metrics continue to align more closely with human preferences. By contrast, the only prior works that optimize over MOS rely on complex multi-step sampling procedures with reinforcement learning [1,2] to approximate human preference alignment using pre-trained neural networks such as MOSNet, whereas our approach can achieve similar goals with a simpler, more generalizable solution through direct optimization.
>
> **Finally, we would like to invite the AC and reviewers to consider this key question:** If our method were as straightforward as suggested, why has no prior work proposed and successfully implemented it, especially given the demonstrated challenges of aligning generative models with human preferences and our framework's incredible effectiveness as acknowledged by all reviewers? We hope the AC and reviewers consider this question when evaluating the novelty and impact of our contributions.
>
> We again thank the reviewer for their comments and hope these clarifications will assist in understanding the meaningful advancements presented in our work.
>
> **References:**
>
> [1] Chen, C., Hu, Y., Wu, W., Wang, H., Chng, E. S., & Zhang, C. (2024). Enhancing Zero-shot Text-to-Speech Synthesis with Human Feedback. arXiv preprint arXiv:2406.00654.
>
> [2] Chen, J., Byun, J. S., Elsner, M., & Perrault, A. (2024). Reinforcement Learning for Fine-tuning Text-to-speech Diffusion Models. arXiv preprint arXiv:2405.14632.

---

### Official Review · Reviewer_i2SD · 2024-11-04

**Soundness:** 4
**Presentation:** 3
**Contribution:** 3
**Rating:** 8
**Confidence:** 4

**Summary:**

The paper presents DMDSpeech, a distilled diffusion model for zero-shot speech synthesis that achieves state-of-the-art performance while significantly reducing inference time. By employing distribution matching distillation, the model generates high-quality speech in just four steps and facilitates direct metric optimization through the use of speaker verification (SV) and connectionist temporal classification (CTC) losses. The authors demonstrate that optimizing these metrics leads to improvements in speaker similarity and word error rate, enhancing the overall intelligibility and quality of synthesized speech. The research highlights the potential of direct metric optimization in bridging the gap between generative modeling and human auditory preferences.

**Strengths:**

**Originality**
- The authors borrow the concept of distribution matching distillation (DMD) from image synthesis. Yet, the use of DMD in speech synthesis using SV and CTC losses appears to be inspiring and original.
- This should be the first successful application of DMD in speech domain that illustrates promising results.

**Quality**
- As far as I checked, the mathematical derivations are technically sound, and I believe some equations are taken from DMD 2.
- The training framework designed for zero-shot TTS is also sensible. Expectedly, the CTC loss improves the text alignment and WER, while the SV loss improves the speaker similarity.

**Clarity**
- Section 3 provides comprehensive details about the model architecture, training procedures, and loss functions used, which can aid readers in reproducing the proposed model.
- The results are presented both subjectively and objectively with comparisons to baseline models, which aids in understanding the effectiveness of the proposed model.
- The use of pitch comparison in figure 2 to confirm the improvement of student model is persuasive. This suggests a decent way to analyze the effects of DMD and direct metric optimization.

**Significance**
- DMDSpeech significantly reduces inference time while achieving competitive performance. This could yield a substantial impact to the industry and relative researches.

**Weaknesses:**

There are some minor issues in this paper.
- In figure 1, the "subtitle" of the four blocks should be consistently placed (e.g. upper leftmost), otherwise the readers may be difficult to spot them. (I just neglected "Inference", which is inconsistently located at the lower part)
- The choice of four steps in DMDSpeech is not discussed well. The readers may wonder why not 3 or 5 or 20 steps, which both can exhibit significant speedups when compared to 128 steps.

**Questions:**

Would the quality be better if we use more steps for the student model?

---

> ### Author Response · Authors · 2024-11-16
> **Point-to-Point Rebuttal to Reviewer i2SD**
>
> We thank the reviewer for appreciating our work, acknowledging its originality and novelty, clarity in presentation and significance it provides to the industry and relevant research. Below are our responses to your questions.
>
> 1. For the subtitle of Fig 1., we will make sure all four subfigures have the same location in the subtitle. We will make this change in the revised submission.
>
> 2. The reason we chose 4 steps is the original DMD 2 paper uses 4 steps, and coincidentally this is also the minimal step number for the state of the art performance. We agree that any step number lower than the teacher can improve the inference speed, but from our experiments we found 4 steps to be the minimum required to obtain the best performance. We will add a discussion about this in the revised version of our paper.

---

### Author Response · Authors · 2024-11-16
**General Response to All Reviewers**

We thank all reviewers for highlighting the key strengths of our work, including its technical soundness, comprehensive experiments, and strong implementation details that ensure reproducibility. Reviewer 4Zge appreciated our effective analysis of mode shrinkage and robust experimental results, while Reviewer 9zdq acknowledged the significant improvements in naturalness, speaker similarity, and inference speed. Reviewer i2SD recognized the originality and significance of our work, specifically highlighting DMDSpeech as the first successful application of distribution matching distillation (DMD) in the speech synthesis domain with SV and CTC loss, achieving state-of-the-art results while reducing inference time. These consistent acknowledgments underscore the rigor, innovation, and practical impact of our contributions to the TTS field.

Regarding the novelty concerns raised, we would like to clarify the distinct contributions of our work:

- **Beyond DMD in Speech Synthesis:**
While distillation has been applied to diffusion models, prior works in speech synthesis (e.g., [1,2,3]) have consistently shown that student models underperform or at best perform on par with teacher models in key metrics such as MOS and SIM. Our work is the first to demonstrate that, through direct metric optimization, the distilled student can consistently outperform the teacher across all standard metrics (MOS, SMOS, WER, SIM) while maintaining faster inference.

-  **Challenges in Direct Metric Optimization:**
Direct metric optimization in TTS is non-trivial in all previous approaches:

   1. **Traditional Models:** Models relying on components like monotonic alignment search (MAS) and duration predictors (such as YourTTS and NaturalSpeech 2&3) cannot propagate gradients to all components such as text encoder and duration predictor, preventing true end-to-end optimization. Although YourTTS uses a speaker similarity loss, this architectural limitation prevents optimization of metrics like WER, which depend heavily on optimizing text representations (text encoder) and phoneme durations (duration predictor).

   2. **Iterative Sampling Models:** Modern transformer-based models (autoregressive or diffusion) require iterative sampling (e.g., 120 steps for autoregressive to generate 3-second audio or 16 steps for most advanced flow-matching models [4]), making backpropagation computationally prohibitive because of memory constraints and unstable due to vanishing gradients.

      Furthermore, it is not possible to directly optimize through the teacher model because reasonably intelligible speech can only be obtained at low noise levels. At higher noise levels, the generated samples are unintelligible and unsuitable for optimization using SV and CTC loss, as the gradients are not instructive enough. We have newly added samples to our demo page to illustrate this limitation, showing that direct optimization through the teacher model is not trivial. These examples highlight why the only viable way to perform direct metric optimization in diffusion models is through distillation. You can find these samples on our demo page: https://dmdspeech.github.io/demo#rebuttal.

- **Our Solution:**
By distilling the diffusion model into a single-step generator, we enable efficient and stable backpropagation for optimizing differentiable metrics. Moreover, our framework generalizes beyond speaker similarity (SIM) and can be applied to word error rate (WER), showcasing it can be extended to any differentiable metrics, offering adaptability as new metrics emerge that can better align with human preference. We believe this represents a meaningful advancement in TTS technology.

**References:**

[1] Ye, Z., Xue, W., Tan, X., Chen, J., Liu, Q., & Guo, Y. (2023, October). Comospeech: One-step speech and singing voice synthesis via consistency model. In Proceedings of the 31st ACM International Conference on Multimedia (pp. 1831-1839).

[2] Ye, Z., Ju, Z., Liu, H., Tan, X., Chen, J., Lu, Y., ... & Guo, Y. (2024, October). Flashspeech: Efficient zero-shot speech synthesis. In Proceedings of the 32nd ACM International Conference on Multimedia (pp. 6998-7007).

[3] Liu, Z., Wang, S., Inoue, S., Bai, Q., & Li, H. (2024). Autoregressive Diffusion Transformer for Text-to-Speech Synthesis. arXiv preprint arXiv:2406.05551.

[4] Chen, Y., Niu, Z., Ma, Z., Deng, K., Wang, C., Zhao, J., ... & Chen, X. (2024). F5-TTS: A Fairytaler that Fakes Fluent and Faithful Speech with Flow Matching. arXiv preprint arXiv:2410.06885.

---

### Note · Authors · 2025-01-31

I have read and agree with the venue's withdrawal policy on behalf of myself and my co-authors.